# LOST DOMAIN GENERALIZATION IS A NATURAL CONSEQUENCE OF LACK OF TRAINING DOMAINS

## ABSTRACT

We show a hardness result for the number of training domains required to achieve a small population error in the test domain. Although many domain generalization algorithms have been developed under various domain-invariance assumptions, there is significant evidence to indicate that out-of-distribution (o.o.d.) test accuracy of state-of-the-art o.o.d. algorithms is on par with empirical risk minimization and random guess on the domain generalization benchmarks such as DomainBed. In this work, we analyze its cause and attribute the lost domain generalization to the lack of training domains. We show that, in a minimax lower bound fashion, *any* learning algorithm that outputs a classifier with an $\epsilon$ excess error to the Bayes optimal classifier requires at least $\mathrm{poly}(1/\epsilon)$ number of training domains, even though the number of training data sampled from each training domain is large. Experiments on the DomainBed benchmark demonstrate that o.o.d. test accuracy is monotonically increasing as the number of training domains increases. Our result sheds light on the intrinsic hardness of domain generalization and suggests benchmarking o.o.d. algorithms by the datasets with a sufficient number of training domains.

## 1 INTRODUCTION

Domain generalization (Mahajan et al., 2021; Dou et al., 2019; Yang et al., 2021; Bui et al., 2021; Robey et al., 2021; Wald et al., 2021; Recht et al., 2019)—where the training distribution is different from the test distribution—has been a central research topic in machine learning (Blanchard et al., 2021; Chuang et al., 2020; Zhou et al., 2021), computer vision (Piratla et al., 2020; Gan et al., 2016; Huang et al., 2021; Song et al., 2019; Taori et al., 2020), and natural language processing (Wang et al., 2021; Fried et al., 2019). In machine learning, the study of domain generalization has led to significant advances in the development of new algorithms for out-of-distribution (o.o.d.) generalization (Li et al., 2022b; Bitterwolf et al., 2022; Thulasidasan et al., 2021). In computer vision and natural language processing, new benchmarks such as DomainBed (Gulrajani & Lopez-Paz, 2021) and WILDs (Koh et al., 2021; Sagawa et al., 2021) are built toward closing the gap between the developed methodology and real-world deployment. In both cases, the problem can be stated as given a set of training domains $\{P_e\}_{e=1}^{E}$ which are drawn from a domain distribution $\mathcal{P}$ and given a set of training data $\{(\mathbf{x}_i^e, y_i^e)\}_{i=1}^{n}$ which are drawn from $P_e$, the goal is to develop an algorithm based on the training data and their domain labels $e$ so that the algorithm in expectation performs well on the unseen test domains drawn from $\mathcal{P}$.

Despite progress on the domain generalization, many fundamental questions remain unresolved. For example, in search of lost domain generalization, Gulrajani & Lopez-Paz (2021) conducted extensive experiments using DomainBed and found that, when carefully implemented, empirical risk minimization (ERM) shows state-of-the-art performance across all datasets despite many algorithms are carefully designed for the out-of-distribution tasks. For example, when the algorithm is trained on the "+90%"[1] and "+80%" domains of the ColoredMNIST dataset (Arjovsky et al., 2019) and is tested on the "−90%" domain, the best-known o.o.d. algorithm achieves test accuracy no better than a random-guess algorithm under all three model selection methods in Gulrajani & Lopez-Paz (2021). Thus, it is natural to ask what causes the lost domain generalization and how to find it?

---

[1] The number refers to the degree of correlation between color and label.

Table 1: The number of domains in the o.o.d. benchmarks WILDs (Koh et al., 2021; Sagawa et al., 2021) and DomainBed (Gulrajani & Lopez-Paz, 2021). It shows that most of the datasets in the two benchmarks suffer from small number of domains, which might not be sufficient to learn a classifier with good domain generalization.

| WILDs | | iWildCam | Camelyon17 | RxRx1 | OGB-MolPCBA | BlobalWheat | CicilComments | FMoW | PovertyMap | Amazon | Py150 |
|---|---|---|---|---|---|---|---|---|---|---|---|
| # domains | | 323 | 5 | 51 | 120,084 | 47 | 16 | 80 | 46 | 2,586 | 8,421 |

| | | DomainBed | | CMNIST | RMNIST | VLCS | PACS | Office-Home | Terra Incognita | DomainNet | | |
|---|---|---|---|---|---|---|---|---|---|---|---|---|
| | | # domains | | 3 | 6 | 4 | 4 | 4 | 4 | 6 | | |

In this paper, we attribute the lost domain generalization to the lack of training domains. Our study is motivated by an observation that off-the-shelf benchmarks often suffer from few training domains. For example, the number of training domains in DomainBed (Gulrajani & Lopez-Paz, 2021) for all its 7 datasets is at most 6; in WILDs (Koh et al., 2021; Sagawa et al., 2021), 7 out of 10 datasets have the number of training domains fewer than 350 (see Table 1). Therefore, one may conjecture that increasing the number of training domains might improve the empirical performance of existing domain generalization algorithms significantly. In this paper, we show that, information theoretically, one requires at least $\text{poly}(1/\epsilon^2)$ number of training domains in order to achieve a small excess error $\epsilon$ for any learning algorithm. This is in sharp contrast to many existing benchmarks in which the number of training domains is limited.

## 2 RELATED WORK

Out-of-distribution (o.o.d) generalization (Hendrycks & Dietterich, 2019; Shankar et al., 2018; Zhou et al., 2021) has received extensive attention in recent years. One representative way is the causal modelling inspired by Invariant Risk Minimization (IRM) (Arjovsky et al., 2019). IRM tries to learn an invariant feature representation to capture the underlying causal mechanism of interest across domains such that the classifier based on this invariant feature representation shall be invariant across all domains. Given multiple training domains, IRM learns invariant representations approximately by adding a regularization. The results of IRM indicate that failing to generalize to o.o.d. data comes from failing to capture the causal factors of variation in different domains. Following IRM, Risk Extrapolation (REx) (Krueger et al., 2021) proposes to reduce differences in risk across training domains. Derivative Invariant Risk Minimization (DIRM) (Bellot & van der Schaar, 2020) maintains the invariance of the gradient of training risks across different domains.

Another line of research uses different metrics to tackle the o.o.d problem. For example, Maximum Mean Discrepancy-Adversarial AutoEncoder (Li et al., 2018b) employs Generative Adversarial Networks and the maximum mean discrepancy metric (Gretton et al., 2012) to align different feature distributions. Mixture of Multiple Latent Domains (Matsuura & Harada, 2020) learns domain-invariant features by clustering techniques without knowing which domain the training samples belong to. Recently, Meta-Learning Domain generalization (Li et al., 2020) employs a lifelong learning method to tackle the sequential problem of new incoming domains.

To explore the o.o.d problem, one line of research focuses on the case where only one training domain is accessible. Causal Semantic Generative model (CSG) (Liu et al., 2021) uses two sets of correlated latent variables, *i.e.*, the semantic and non-semantic features, to model the relation between the data and the corresponding labels. In their assumption, the semantic features relate the data to their corresponding labels while the non-semantic features only affect the generation of data. CSG decouples the semantic and non-semantic features to improve o.o.d generalization given only one training domain.

However, recent work (Gulrajani & Lopez-Paz, 2021) claims that all existing algorithms cannot capture the true invariant feature and observes that their performance is on par with ERM and random guess on several datasets. In this paper, to explain why it occurs, we theoretically analyze the o.o.d. generalization problem and provide a minimax lower bound for the number of training domains required to achieve a small population error in the test domain. Massart & Nédélec (2006a) have proved that it requires at least $\Omega(1/\epsilon^2)$ samples from a distribution to estimate the success probability of a Bernoulli variable with an $\epsilon$ error. Motivated by this, we observe a similar phenomenon and prove that the learning algorithms need at least $\Omega(1/\epsilon^2)$ number of training domains. Recently, a

concurrent work (Li et al., 2022a) presents an upper bound on the expected excess error of the ERM algorithm using the Rademacher complexity. Similarly, another work (Blanchard et al., 2021) gives an upper bound on the excess error of general learning algorithms with high probability and shows that the sample size of each domain is inversely proportional to the excess error. On the other side, while previous work (Li et al., 2022a; Blanchard et al., 2021) showed positive results on the domain generalization, we present a negative result (*i.e.*, a lower bound regarding the number of training domains) on the expected excess error for all possible learning algorithms.

## 3 MINIMAX LOWER BOUND FOR DOMAIN GENERALIZATION

In this section, we provide a minimax lower bound for domain generalization. Our results lower bound the number of training domains required for good o.o.d. generalization.

**Notation.** We will use *bold capital* letters such as $\mathbf{X}$ to represent a random vector, *bold lower-case* letters such as $\mathbf{x}$ to represent the implementation of a random vector, capital letters such as $Y$ to represent a random variable, and lower-case letters such as $y$ to represent the implementation of a random variable. Specifically, we denote by $\mathbf{X}$ the random vector of instance, denote by $\mathbf{x}$ the implementation of random vector $\mathbf{X}$, denote by $Y$ the random variable of label, and denote by $y \in \{0, 1\}$ the implementation of random variable $Y$. We will use $L(f)$ to represent the expected 0-1 loss of classifier $f$ w.r.t. the mixture of data distributions of all domains, *i.e.*, $L(f) = \Pr_{(\mathbf{X},Y)}(f(\mathbf{X}) \neq Y)$. Throughout the paper, we will frequently use $\mathcal{P}$ to represent the distribution of distribution, *i.e.*, the domain distribution, will use $P_e$ to represent the data distribution of the $e$-th domain, and will use $(\mathbf{x}^e, y^e)$ to represent the data sampled from the $e$-th domain $P_e$. We call $e \in \{1, 2, 3, ...\}$ the domain labels, which are accessible to the learner.

**Problem setups.** In our hard instance, we view the $e$-th domain as a data distribution $P_e$ given by $\Pr(\mathbf{X}, Y | \mathbf{B}_e = \mathbf{b}_e)$, where $e$ is the domain label and $\mathbf{B}_e$'s represent i.i.d. Bernoulli random vectors that parameterize the data distribution of the $e$-th domain. In this paper, we will regard $\Pr(\mathbf{X}, Y | \mathbf{B}_{e_1})$ and $\Pr(\mathbf{X}, Y | \mathbf{B}_{e_2})$ as two different domains as long as $e_1 \neq e_2$. We assume that each domain is sampled from a domain distribution $\mathcal{P}$ (*i.e.*, the distribution of $\mathbf{B}_e$), and the data in the $e$-th domain are sampled from a data distribution $P_e$ given by $\Pr(\mathbf{X}, Y | \mathbf{B}_e = \mathbf{b}_e)$. Let $f^*$ be the Bayes optimal classifier of the mixture of data distributions across all domains, and assume $f^* \in \mathcal{F}$, where $\mathcal{F}$ can be any function class such as deep neural networks. For any $h \in [0, 1]$, we define a class of domain distributions by $\mathcal{P}(h, \mathcal{F}) := \{\mathcal{P} : |2 \Pr(\mathbf{B}_e = 1) - 1| \geq h\}$. Note that the margin parameter $h$ determines the randomness of the domain: large $h$ (e.g., $h = 1$) means $\Pr(\mathbf{B}_e = 1)$ is bounded away from $1/2$. We will investigate the following minimax risk:

$$R_{E,n}(h, \mathcal{F}) := \inf_{\widetilde{f}_{E,n} \in \mathcal{F}} \sup_{\mathcal{P}} \mathbf{E}_{P_e \sim \mathcal{P}} \mathbf{E}_{(\mathbf{X}^e, Y^e) \sim P_e} \left[ L(\widetilde{f}_{E,n}) - L(f^*) \right], \tag{1}$$

where $E$ is the number of training domains, $n$ is the number of training samples from each domain, and the two expectations are taken over the sampling of training data and domains to learn $\widetilde{f}_{E,n}$. The minimax problem in Equation (1) characterizes the access risk of the best learning algorithm with an access to $E$ training domains and $n$ data samples under the worst-case domain distribution.

Let $V$ be the VC dimension of $\mathcal{F}$, which is defined as the maximum number of points that can be arranged so that $\mathcal{F}$ shatters them. Our main results are as follows:

**Theorem 1.** *For $n = \infty$, any $h \in [0, 1]$ and any $E \geq V$, we have the lower bound*

$$R_{E,\infty}(h, \mathcal{F}) \geq c \min \left( \frac{V - 1}{Eh}, \sqrt{\frac{V - 1}{E}} \right), \tag{2}$$

*where $c > 0$ is an absolute constant.*

We defer the proofs of Theorem 1 to Appendix A. The theorem provides a lower bound on the number of training domains required to achieve a small population error, even though one can sample as many data points as possible from each domain. The case of $n = \infty$ captures the "easiest" case for the learner, where the learning algorithm can access to full knowledge about each training domain. The case of finite $n$ is harder than $n = \infty$, as the learner has only partial knowledge about each training domain and $R_{E,n}(h, \mathcal{F}) \geq R_{E,\infty}(h, \mathcal{F})$. Therefore, *Equation (2) provides a universal lower bound*

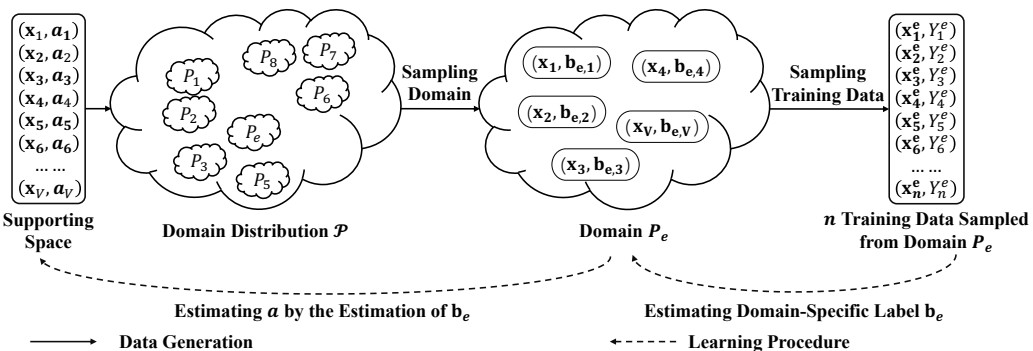

Figure 1: Illustration of our o.o.d. generalization problem. We show how data is sampled and how learning algorithms learn the knowledge. When generating training data, $E$ domain from the data distribution $\mathcal{P}$ are first sampled, and after that for each domain, $n$ training data are sampled to form the training dataset which will be fed into a learning algorithm. The learning algorithm recovers the underlying label $\mathbf{a}$ by the estimation of the underlying label $\mathbf{b}_e, e \in [E]$ under the observation of training data.

*for general $n \geq 1$.* Theorem 1 implies that, information-theoretically, one requires at least $\mathrm{poly}(V/\epsilon)$ number of training domains in order to achieve a small excess error $\epsilon$ for any learning algorithm of $\mathcal{F}$. This is in sharp contrast to many existing benchmarks on which the number of training domains is limited (see Table 1). For example, in the celebrated ColoredMNIST dataset (Arjovsky et al., 2019), there are only 2 training domains. When the algorithm is trained on the "$+90\%$" and "$+80\%$" domains and is tested on the "$-90\%$" domain, the best-known o.o.d. algorithm achieves test accuracy no better than random guess under all three model selection methods in Gulrajani & Lopez-Paz (2021). Theorem 1 predicts the failures of future algorithms on these datasets and attributes the poor performance of existing o.o.d. algorithms to the lack of training domains.

**Differences between our work and Massart & Nédélec (2006b).** The major differences include: 1) the construction of the hard instance, *i.e.*, a two-stage data generative procedure, and 2) the strategy of splitting the hard problem into two sub-problems (see Figure 1). These two aspects are original and separate our contributions with previous works. For 1), our data generative model first samples $E$ domains from the domain distribution $\mathcal{P}$ by generating the domain-specific label $\mathbf{b}_e, \forall e \in [E]$ and then samples the training data from each sampled domain. On the other hand, Massart & Nédélec (2006b) considered a totally different scenario: they investigated the effect of training sample size on the excess risk in the single-domain problem when the training and test data are i.i.d. For 2), our proof has to deal with two expectations given that we have designed a novel two-stage recovery strategy. Our two-stage problem splits the hard problem into two simpler problems which estimate a binary string $\mathbf{a}$ and $\mathbf{b}_e, \forall e \in [E]$, while Massart & Nédélec (2006b) only considered one binary string estimation problem. The two binary string estimation problems are entangled, making our analysis more challenging.

## 4 EXPERIMENTS

Theorem 1 shows that *any* learning algorithm that outputs a classifier with an $\epsilon$ excess error to the Bayes optimal classifier requires at least $\mathrm{poly}(1/\epsilon)$ number of training domains, even though the number of training data sampled from each training domain is large. In this section, we complement our theoretical results with an empirical study to evaluate the impact of number of training domains.

### 4.1 DATASETS

We conducted extensive experiments on two datasets from DomainBed, *i.e.*, ColoredMNIST (Arjovsky et al., 2019) and RotatedMNIST (Ghifary et al., 2015). We notice that there are other popular domain generalization datasets, *e.g.*, PACS (Li et al., 2017), VLCS (Fang et al., 2013), Office-

Home (Venkateswara et al., 2017), and Terra Incognita (Beery et al., 2018). However, these datasets are hard to generate more training domains synthetically as their data generation process cannot be parameterized by a single variable (*e.g.*, correlation between color and label in ColoredMNIST, or rotation degree in RotatedMNIST). Thus, in our paper we do not consider these datasets.

**ColoredMNIST** (Arjovsky et al., 2019) is a variant of the MNIST hand written digit classification dataset (LeCun et al., 1998). It is a synthetic dataset containing three domains $p_e \in [0.1, 0.2, 0.9]$ colored either red or blue formalizing $70,000$ examples of dimension $(2, 28, 28)$ and 2 classes. The label is a noisy function of the digit and color, such that color bears correlation $p_E$ with the label and the digit bears correlation $0.75$ with the label. Inspired by the protocol introduced in DomainBed repository, we randomly split the training dataset into 10 subdatasets with equal training samples. Each domain of ColoredMNIST is generated as follows: 1) Assign a preliminary binary label $y'$ to the image based on the digit: $y' = 0$ for digits $0 - 4$ and $y' = 1$ for $5 - 9$; 2) Obtain the final label $y$ by flipping $y'$ with probability $0.25$; 3) Sample the color id $z$ by flipping $y$ with probability $p_e$; 4) Color the image red if $z = 1$ or green if $z = 0$. The only parameter of a training domain is $p_e$. We use the domain with $p_e = 0.5$ as the test domain and uniformly sample $E$ parameters $p_e$ from $(0, 1)/\{0.5\}$ to form $E$ training domains. Each domain randomly uses a subset from 10 subdatasets of the whole MNIST to generate the data.

**RotatedMNIST** (Ghifary et al., 2015) is another variant of MNIST with 6 domains containing digits rotated by $\{0, 15, 30, 45, 60, 75\}$ degrees. It contains $70,000$ examples of dimension $(1, 28, 28)$ and 10 classes. Similar to ColoredMNIST, we use the domain with $45$ degrees rotation as the test domain and uniformly sample $E$ rotation degrees from $[0, 90)/\{45\}$ to form $E$ training domains.

## 4.2 Algorithms

To validate our theoretical results, we evaluate the effect of number of training domains on o.o.d. algorithms, including ERM (Empirical Risk Minimization) (Vapnik, 1991), IRM (Invariant Risk Minimization) (Arjovsky et al., 2019), GroupDRO (Sagawa et al., 2020), Mixup (Xu et al., 2020), MLDG (Li et al., 2018a), CORAL (Sun & Saenko, 2016), MMD (Li et al., 2018b), DANN (Ganin et al., 2016), and C-DANN (Li et al., 2018c). The details of the algorithms are shown in the appendix.

For each algorithm, we employ the default hyper-parameter introduced in Section D.2 of DomainBed (Gulrajani & Lopez-Paz, 2021), as our goal is not to show the best performance of algorithms but to show the correlations to our theoretical results. Following DomainBed (Gulrajani & Lopez-Paz, 2021), we use MUNIT (Table 4) for ColoredMNIST and RotatedMNIST.

## 4.3 Evaluation settings

We train models using 9 different Domain Generalization algorithms, with a varying number of training domains on ColoredMNIST and RotatedMNIST. Each trial is done with 5 different random seeds, and we present the average results. We use the code repository of DomainBed (Gulrajani & Lopez-Paz, 2021) with PyTorch (Paszke et al., 2019).

**Model Evaluation.** Following DomainBed (Gulrajani & Lopez-Paz, 2021), we employ and adapt three different model selection methods for training algorithms as shown below,

- **Leave-one-domain-out cross-validation.** $E$ models are trained on $E$ training domains with equal hyperparameters, while each experiment holds out one of the training domains. The evaluation of each model is conducted on its held-out domain and we choose the model when maximizing the accuracy on the held-out domain. This method has an assumption that training and test domains are drawn from a *meta-distribution over domains*, and that our goal is to maximize the expected performance under this meta-distribution. This method corresponds our data generation model. But it requires huge computational resources. We only use this method to select model with the number of domains varying from 2 to 30.

- **Training-domain validation set.** Each training domain is split into training and validation subsets and the overall validation set consists of the validation subsets of each training domain. Finally, we choose the model maximizing the accuracy on the overall validation set. This method has an assumption that the training and test examples follow *similar distributions*.

Table 2: The experimental results on **ColoredMNIST** with ERM, IRM, GroupDRO, Mixup, MLDG, and CORAL w.r.t the number of training domain using the training-domain validation set model selection method.

| \# | ERM | IRM | GroupDRO | Mixup | MLDG | CORAL |
|---|---|---|---|---|---|---|
| 4 | 0.6697±0.0120 | 0.5500±0.0091 | 0.6710±0.0134 | 0.6081±0.0144 | 0.6744±0.0046 | 0.6586±0.0139 |
| 6 | 0.7135±0.0027 | 0.5910±0.0072 | 0.7158±0.0013 | 0.6703±0.0088 | 0.7107±0.0035 | 0.7141±0.0020 |
| 8 | 0.7183±0.0018 | 0.6278±0.0031 | 0.7199±0.0016 | 0.7129±0.0017 | 0.7195±0.0005 | 0.7199±0.0013 |
| 10 | 0.7226±0.0005 | 0.6685±0.0086 | 0.7220±0.0004 | 0.7159±0.0006 | 0.7271±0.0011 | 0.7228±0.0004 |
| 12 | 0.7280±0.0011 | 0.6968±0.0034 | 0.7288±0.0012 | 0.7223±0.0015 | 0.7287±0.0008 | 0.7278±0.0010 |
| 14 | 0.7289±0.0016 | 0.6709±0.0123 | 0.7284±0.0016 | 0.7215±0.0007 | 0.7316±0.0015 | 0.7285±0.0015 |
| 16 | 0.7268±0.0011 | 0.6777±0.0055 | 0.7272±0.0010 | 0.7230±0.0014 | 0.7322±0.0008 | 0.7258±0.0010 |
| 18 | 0.7304±0.0017 | 0.7031±0.0045 | 0.7292±0.0018 | 0.7255±0.0015 | 0.7338±0.0009 | 0.7297±0.0008 |
| 20 | 0.7305±0.0018 | 0.6957±0.0069 | 0.7321±0.0011 | 0.7239±0.0010 | 0.7336±0.0008 | 0.7311±0.0013 |
| 22 | 0.7323±0.0011 | 0.6935±0.0078 | 0.7298±0.0010 | 0.7276±0.0012 | 0.7368±0.0014 | 0.7296±0.0011 |
| 24 | 0.7330±0.0015 | 0.6908±0.0086 | 0.7358±0.0009 | 0.7269±0.0014 | 0.7366±0.0012 | 0.7354±0.0012 |
| 26 | 0.7350±0.0019 | 0.6995±0.0026 | 0.7343±0.0016 | 0.7323±0.0013 | 0.7366±0.0011 | 0.7353±0.0015 |
| 28 | 0.7336±0.0016 | 0.6997±0.0076 | 0.7347±0.0014 | 0.7327±0.0011 | 0.7370±0.0013 | 0.7332±0.0014 |
| 30 | 0.7331±0.0023 | 0.7113±0.0027 | 0.7326±0.0023 | 0.7297±0.0020 | 0.7391±0.0012 | 0.7329±0.0020 |
| 48 | 0.7386±0.0014 | 0.7219±0.0007 | 0.7398±0.0015 | 0.7352±0.0014 | 0.7410±0.0010 | 0.7385±0.0014 |
| 96 | 0.7427±0.0014 | 0.7182±0.0015 | 0.7424±0.0013 | 0.7399±0.0012 | 0.7444±0.0011 | 0.7424±0.0014 |
| 192 | 0.7437±0.0014 | 0.7287±0.0012 | 0.7443±0.0014 | 0.7424±0.0013 | 0.7461±0.0010 | 0.7437±0.0014 |

- **Test-domain validation set (oracle).** We choose the model maximizing the accuracy on a validation set that follows the distribution of the test domain. All the models are trained for the same fixed number of steps and the final checkpoints are used for evaluation. It assumes and requires that models have the access to the test domain which might not be possible in the real-world application.

## 4.4 EXPERIMENTAL RESULTS ON COLOREDMNIST AND ROTATEDMNIST

We first introduce the average results on two different datasets using 9 algorithms with the number of training domains varying from 2 to 192 and then present the results with limited number of domains. Due to the limitation of space, we present the most important results in our paper while leaving the left results in the Appendix.

### 4.4.1 EVALUATING THE EFFECT OF NUMBER OF TRAINING DOMAINS

**Results.** We run the experiments on ColoredMNIST and RotatedMNIST with ERM, IRM, Group-DRO, Mixup, MLDG, CORAL, MMD, DANN and C-DANN while the number of training domains varies from 2 to 192. The average accuracy w.r.t the number of training domains is shown in Tables 2 and 3 in the main paper, Tables 6, and 8, and Figures 4 and 5 in the Appendix. It shows that the test accuracy is proportional to the number of training domains with all the algorithms on both ColoredMNIST and RotatedMNIST which is consistent with our theoretical results (Theorem 1).

**Training-domain validation set analysis.** The results are shown in Tables 2 and 8 and Figure 5. We observe that the test accuracy of almost all the algorithms on both ColoredMNIST and RotatedMNIST is monotonically increasing as the number of training domains grows while the accuracy of IRM on both datasets, MMD on ColoredMNIST, DANN and CDANN on RotatedMNIST experiences slight drops for certain number of training domains. We also find that the standard deviations of MMD are quite big which might be due to the hyperparameter setting as we did not try to tune the hyperparameters to gain the best performance. Besides, the standard deviations of all the algorithms on the first experiments (the least number of training domains) are quite large. That is because the

Table 3: The experimental results on **ColoredMNIST** with ERM, IRM, GroupDRO, Mixup, MLDG, and CORAL w.r.t the number of training domain using the test-domain validation set (oracle) model selection method.

| \# | ERM | IRM | GroupDRO | Mixup | MLDG | CORAL |
|---|---|---|---|---|---|---|
| 4 | 0.6697±0.0120 | 0.5517±0.0085 | 0.6710±0.0134 | 0.6081±0.0144 | 0.6750±0.0046 | 0.6586±0.0139 |
| 6 | 0.7138±0.0027 | 0.5915±0.0073 | 0.7158±0.0013 | 0.6703±0.0088 | 0.7133±0.0038 | 0.7141±0.0020 |
| 8 | 0.7203±0.0014 | 0.6278±0.0031 | 0.7205±0.0017 | 0.7129±0.0017 | 0.7213±0.0009 | 0.7209±0.0010 |
| 10 | 0.7244±0.0012 | 0.6685±0.0086 | 0.7236±0.0012 | 0.7159±0.0006 | 0.7271±0.0011 | 0.7244±0.0012 |
| 12 | 0.7284±0.0009 | 0.6968±0.0034 | 0.7288±0.0012 | 0.7224±0.0015 | 0.7302±0.0010 | 0.7280±0.0010 |
| 14 | 0.7291±0.0017 | 0.6709±0.0123 | 0.7284±0.0016 | 0.7216±0.0007 | 0.7317±0.0015 | 0.7286±0.0015 |
| 16 | 0.7274±0.0011 | 0.6777±0.0055 | 0.7274±0.0010 | 0.7230±0.0013 | 0.7326±0.0010 | 0.7265±0.0011 |
| 18 | 0.7314±0.0012 | 0.7031±0.0045 | 0.7311±0.0012 | 0.7260±0.0016 | 0.7343±0.0008 | 0.7307±0.0010 |
| 20 | 0.7311±0.0015 | 0.6958±0.0069 | 0.7321±0.0011 | 0.7259±0.0010 | 0.7341±0.0010 | 0.7313±0.0012 |
| 22 | 0.7323±0.0011 | 0.6935±0.0078 | 0.7305±0.0012 | 0.7278±0.0011 | 0.7371±0.0014 | 0.7306±0.0014 |
| 24 | 0.7357±0.0013 | 0.6908±0.0086 | 0.7358±0.0009 | 0.7281±0.0018 | 0.7372±0.0012 | 0.7354±0.0012 |
| 26 | 0.7351±0.0018 | 0.6995±0.0026 | 0.7345±0.0015 | 0.7323±0.0013 | 0.7368±0.0011 | 0.7353±0.0015 |
| 28 | 0.7341±0.0015 | 0.6997±0.0076 | 0.7351±0.0014 | 0.7331±0.0011 | 0.7370±0.0013 | 0.7338±0.0012 |
| 30 | 0.7333±0.0023 | 0.7113±0.0027 | 0.7338±0.0024 | 0.7300±0.0018 | 0.7396±0.0014 | 0.7334±0.0021 |
| 48 | 0.7390±0.0012 | 0.7219±0.0007 | 0.7398±0.0015 | 0.7362±0.0012 | 0.7415±0.0009 | 0.7390±0.0014 |
| 96 | 0.7432±0.0014 | 0.7182±0.0015 | 0.7425±0.0013 | 0.7401±0.0011 | 0.7446±0.0011 | 0.7427±0.0014 |
| 192 | 0.7439±0.0012 | 0.7287±0.0012 | 0.7448±0.0012 | 0.7426±0.0014 | 0.7468±0.0010 | 0.7439±0.0013 |

number of training domains is limited and the algorithms are hard to capture general patterns. As the number of training domains grows, the standard deviations of almost all the algorithms decrease.

**Test-domain validation set (oracle) analysis.** Figure 4 and Tables 3, 6 show the results using the oracle model selection method. Similar observations can be obtained. The accuracy of ERM, DANN, CDANN, CORAL, GroupDRO and Mixup on ColoredMNIST and RotatedMNIST is proportional to the number of training domains while there are fluctuations in the lines of MMD and IRM on both datasets, which might be due to the fact that MMD and IRM are sensitive to the hyperparameters as we did not tune the hyperparameters for the best performance. The line of IRM on RotatedMNIST drops slightly when the number of training domains is over $100$. That might be caused by the limited number of training images $n$ in our theorem. In that case, algorithms might not be able to extract general patterns and might learn biased information, which causes the performance drop. Besides, as we only conduct 5 trials for each experiment, the randomness of the experiments might be also another reason why the performance of IRM on RotatedMNIST drops slightly. Overall, the results using test-domain validation set and training-domain validation set model selection methods are the same, which supports our theoretical results.

### 4.4.2 EVALUATING ON THE LIMITED NUMBER OF TRAINING DOMAINS

As the leave-one-domain-out cross-validation requires huge computational resources, we only conduct the experiments with the number of training domain from 2 to 30 with a step of 2. The results are shown in Figure 2 in the main paper, Figure 6, Table 9 and Table 10 in the Appendix. The reason why we only choose the even numbers is that we are trying to sample the domains evenly. For example, for ColoredMNIST, if we use 3 training domains and test on the $0.5$ domain, we may have to sample either two domains whose hyper-parameters are bigger than $0.5$ or smaller than $0.5$, and only one domain whose hyper-parameter is smaller than $0.5$ or bigger than $0.5$, which might cause domain sampling drift and further we observe biased results.

**Analysis of leave-one-domain-out cross-validation results.** Observed from two tables and the figure, we conclude that the test accuracy of most algorithms is proportional to the number of training domain, while there are some exceptions, *e.g.*, IRM on ColoredMNIST and GroupDRO on ColoredMNIST. For all the results on RotatedMNIST, we observe that the results perfectly match

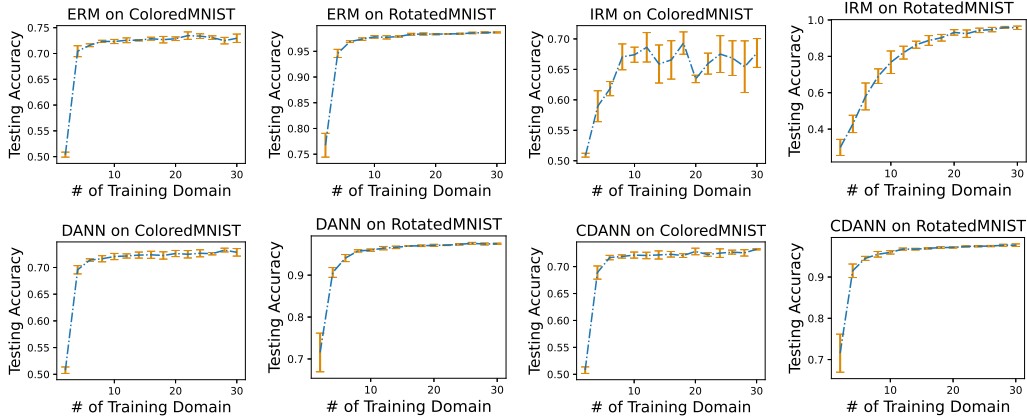

Figure 2: The experimental results on **ColoredMNIST** and **RotatedMINST** using ERM, IRM, DANN and C-DANN w.r.t the number of training domain using the leave-one-domain-out cross-validation method.

our theoretical results even without any hyper-parameter tuning, especially for the experiments on IRM. But we still can observe exactly the same fact that without any hyper-parameter tuning, the test accuracy of IRM on RotatedMNIST grows with the increase of the number of training domain. There are some fluctuations in the line of GroupDRO, CORAL and Mixup, which might be due to the randomness of the experiments as we only conduct 5 experiments for each trial. On the other hand, the results on ColoredMNIST are still proportional to the number of training domains on all algorithms except IRM. And we also observe fluctuation on almost every algorithm. That may be due to the fact that the number of data from each domain, $n$, is $1,000$, which is much smaller than that for RotatedMNIST, which is $10,000$. The limited number of data from each domain might hinder algorithms from learning enough patterns and thus the algorithms can only capture biased patterns due to the lack of training data from each domain. For IRM, similarly, the importance of hyper-parameter tuning causes the accuracy to go up and down with the increase of the number of training domains and one might have to utilize as much as possible computational resources for tuning it. Comparing with other two model selections methods, as this one requires more trials, the standard deviations are smaller.

### 4.5 ABLATION STUDY

**Analysis on different architecture.** To test how our theoretical results generalize to other architecture of neural networks, we further conduct experiments on **ColoredMNIST** (Table 4) with VGG11 (Simonyan & Zisserman, 2014) with the oracle model selection method. We use the learning rate of $5e - 5$ while remaining other hyparameters the same. The corresponding results are shown in Tables 15, 16, 17, and 18 and the left two figures in Figure 3. Similar conclusion can be summarized that the test accuracy still grows with the increase of number of training domain while using a total different architecture. But we observe more fluctuation in the line with VGG11 as we only conduct 1 trail for VGG11. There is a big "valley" around $E = 100$ in the experiments of ERM on **ColoredMNIST**, which is quite unusual as it is too big compared with other fluctuation. That might be caused by the randomness or the failure of hardware as we only see that situation once. We also conducted experiments on Resnet18 (He et al., 2016). But after we tried different set of hyper-parameters, ResNet18 seems not to converge on any set of hyperparamters.

**Analysis on the number $n$ of data from each domain.** To testify the effect of the number of data from each domain, we conduct experiments on **ColoredMNIST** using ERM and IRM with $n$ from 1000 to 20000 and the oracle model selection method, while the original $n$ is set to be 7000. The experimental results are shown in Tables 11, 12, 13, and 14 and the right two figures in Figure 3. It shows that when $n$ is relatively small compared with the original number 7000, especially when $n = 1000$, the line of the accuracy experiences lots of fluctuations. The randomness may be the biggest reason as we only conducted one trail for the ablation study, while we still observe that, the

Figure 3: The experimental results on **ColoredMNIST** using ERM and IRM, w.r.t the number of training domain with the oracle model selection method. The left two figures show the results with different architectures, *i.e.*, MUNIT and VGG11 (Simonyan & Zisserman, 2014), while the left three figures present the corresponding results with different number of $n$.

test accuracy is "overall" proportional to the number of training domain. When $n \geq 2000$, the line of test accuracy is absolutely proportional to the number of training domain, which fits our theoretical results well. But that also arises a question that, what is the minimal requirement on $n$ for achieving a similar theoretical result.

**Discussion on $E < V$.** When $E < V$, there would be a lot of domains that the learning algorithm has never seen in the training phase. Under this assumption, the lower bound on the excess error might be higher than the current results (Theorem 1). But we might be able to have the similar conclusion with our theoretical result. The experimental results shown in Table 9, Table 10 and Figure 2 indicate that, even when the number of domains (less than 30) is relatively small compared with the dimension of the training data in the case $E < V$, the performance is still proportional to the number of training domain $E$ in the most of cases, which supports our theoretical results (Theorem 1).

## 5 CONCLUSION

In this paper, we investigated the out-of-distribution problem and analyzed how many training domains were required to achieve a small population error in the test domain under reasonable assumptions. Our results theoretically characterized the phenomenon of the lost domain generalization which had been found by Gulrajani & Lopez-Paz (2021) in 2021. And our work showed that in a minimax lower bound fashion, *any* learning algorithm with an $\epsilon$ excess error to the Bayes optimal classifier required at least $\mathrm{poly}(1/\epsilon)$ number of training domains, even when the number of training data sampled from each training domain was large. There were strong correlations between our work and some empirical results (Arjovsky et al., 2019; Liu et al., 2021; Krueger et al., 2021) in the o.o.d area. Besides, though we used Bernoulli (discrete) random variables to present our theoretical results, our lower bounds hold true for broader distribution class as we look at the worst-case distributions.

To complement our theoretical results, we conduct experiments on two OOD benchmarks, *i.e.*, ColoredMNIST and RotatedMNIST, with several OOD methods, showing that for the methods used in this paper, the test accuracy on the test domain was proportional to the number of training domains under three different model selection methods. That matched our theoretical results perfectly.

There are several future directions of our work. Our theorem assumed that the number of data samples $n$ from each domain was $\infty$. This assumption was used to lower bound the case of general $n$ because intuitively, the case of $n = \infty$ should be simpler than the case where $n$ is a finite number. It is interesting to understand how $n$ affects a tight minimax lower bound. Another future direction is to explore the case where the numbers of samples from each domain are different. It would be interesting to see which domain dominates the training procedure and how to design o.o.d training algorithms under this scenario. Moreover, in our case, the instance support (feature space) was shared across domains. Another case we should consider is that each domain only has its own instance support. This kind of domain shift is frequently observed in real-world scenarios and it would help us understand the o.o.d problem further. Besides, we would also like to explore the upper bound of o.o.d problems to see whether our lower bound results match the upper bound. Last, though multi-class classification can be seen as a combination of multiple binary classification problems (*e.g.*, one-vs.-rest classifier), it is interesting to extend our results to the multi-classification problem.

ETHICS STATEMENT

We did not see obvious negative ethical impacts in our work. In contrast, our work might have a positive impact on society regarding the fairness (Barocas et al., 2019) and security (Zhang et al., 2019; Kawaguchi et al., 2017; Bubeck et al., 2020) of machine learning. Achieving a small population error in the test domain ensures fairness regarding the bias of the dataset in the race, gender, age, etc., as most of the public datasets, *e.g.*, CelebA (Liu et al., 2015), have bias and that will lead to the bias of machine learning models (Barocas et al., 2019). Moreover, security issues such as adversarial robustness (Hendrycks et al., 2020; 2021) are also related to our study of domain generalization, where clean examples and adversarial examples are from different domains. Improving the population error of machine learning models in the test domain may lead to robust models against adversarial attacks.

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

# A PROOF OF THEOREM 1

## A.1 PREPARATION

We first construct a "hard" instance over the supremum to lower bound the minimax problem and then reduce it into a label recovery problem. We begin with the definition of domain distribution and some lemmas, and we assume $h > 0$ throughout our proof.

**Definition 2.** *A domain distribution is said to satisfy the Massart noise condition with margin $h$ if $|2\eta(\mathbf{X}) - 1| \geq h$ with probability 1. We denote the set of domain distributions satisfying the Massart noise condition by $\mathcal{P}(h, \mathcal{F}) = \{P \in \mathcal{P}(h, \mathcal{F}) : |2\eta(\mathbf{X}) - 1| \geq h \text{ with probability } 1\}$.*

**Lemma 3.** *For any classifier $f : \mathcal{X} \to \{0, 1\}$, any distribution $P$ on $\mathcal{X} \times \{0, 1\}$ and the Bayes optimal classifier $f^*$ on $P$, we have*

$$L(f) - L(f^*) = \mathbf{E}|2\eta(\mathbf{X}) - 1||f(\mathbf{X}) - f^*(\mathbf{X})|.$$

*Specifically, if $P \in \mathcal{P}(h, \mathcal{F})$, we have*

$$L(f) - L(f^*) \geq h\mathbf{E}|f(\mathbf{X}) - f^*(\mathbf{X})| = h\|f - f^*\|_{L_1},$$

*where the $L_1$ norm is computed w.r.t the distribution of $\mathbf{X}$, i.e., $\|f - f^*\|_{L_1} = \sum_{\mathbf{x} \in \mathbf{X}} \Pr(\mathbf{X} = \mathbf{x})|f(\mathbf{x}) - f^*(\mathbf{x})|$ if $\mathbf{X}$ is drawn from a discrete distribution or $\|f - f^*\|_{L_1} = \int_{\mathbf{x} \in \mathbf{X}} p(\mathbf{x})|f(\mathbf{x}) - f^*(\mathbf{x})|d\mathbf{x}$ if $\mathbf{X}$ is drawn from a continuous distribution.*

**Lemma 4.** *Let $f$ be any learning algorithm satisfying $f \in \mathcal{F}$, $\tilde{f}$ be the learning algorithm which is closest to $f^*_\beta$ in $L_1$ norm w.r.t the distribution of sampled data $\mathbf{x}$, i.e., $\tilde{f} = \operatorname{argmin}_{f \in \mathcal{F}} \|f - f^*_\beta\|_{L_1}$, where $f^*_\beta$ is a classifier given by a binary string $\beta$ on $\mathbf{x}_1, \mathbf{x}_2, ..., \mathbf{x}_{V-1}$ such that $\beta = [f^*_\beta(\mathbf{x}_1), \ldots, f^*_\beta(\mathbf{x}_{V-1})]$. For any $\beta \in \{0, 1\}^{V-1}$, we have*

$$\|\tilde{f} - f^*_\beta\|_{L_1} \leq 2\|f - f^*_\beta\|_{L_1}.$$

The goal of $\tilde{f}_{E,n}$ is to estimate the ground-truth label $\mathbf{a}$ of the supporting data by observing and learning from $E \cdot n$ data points sampled from $E$ domains. We observe that, given any domain distribution $\mathcal{D} \in \mathcal{P}(h, \mathcal{F})$, the minimax risk $R_{E,n}(h, \mathcal{F})$ (1) can be lower-bounded as

$$R_{E,n}(h, \mathcal{F}) \geq \inf_{\tilde{f}_{E,n} \in \mathcal{F}} \sup_{\mathcal{D} \in \mathcal{P}(h, \mathcal{F})} \mathbf{E}_{P_e \sim \mathcal{D}} \mathbf{E}_{(\mathbf{X}^e, Y^e) \sim P_e} \left[ L(\tilde{f}_{E,n}) - L(f^*) \right]. \tag{3}$$

$\mathcal{D}$ will be constructed such that the supporting space $(\mathbf{X}, \mathbf{a})$ of $\mathcal{D}$ contains $V - 1$ data points $(\mathbf{x}_i, a_i), i \in [V - 1]$. We denote by $\{\mathbf{x}_i\}_{i \in [V-1]}$ and $\{a_i\}_{i \in [V-1]}$ the feature space and the label space, respectively. The label space of $\mathcal{D}$ can be indexed by the vertices of a binary hypercube and the expected excess risk can be reduced to the problem of label recovering. The feature support $\mathbf{x}_1, \mathbf{x}_2, ..., \mathbf{x}_V$ is shared by all the domains in $\mathcal{D}$. All learning algorithms $f$ with the VC dimension bigger than $V$ can shatter these data points.

**Theorem 5.** *Let $\{P_\theta : \theta \in \{0, 1\}^m\}$ be a collection of probability distributions on some set $Z$ index by the vertices of the binary hypercube $\Theta = \{0, 1\}^m$. Suppose that there exists some constant $\alpha > 0$, such that*

$$H^2(P_\theta, P_{\theta'}) \leq \alpha, \quad \text{if } d_H(\theta, \theta') = 1. \tag{4}$$

*Consider the problem of estimating the parameter $\theta \in \Theta$ based on $n$ i.i.d observations from $P_\theta$, where the loss is measured by the Hamming distance. Then the corresponding minimax risk, which we denote by $\mathcal{M}_n(\Theta)$, is lower-bounded as*

$$\mathcal{M}_n(\Theta) \geq \frac{m}{2}(1 - \sqrt{\alpha n}). \tag{5}$$

## A.2 CONSTRUCTING HARD INSTANCE

In this part, we lower bound this minimax problem by constructing a "hard" domain distribution $\mathcal{D} \in \mathcal{P}(h, \mathcal{F})$. We first show how to sample $E$ training domains from the domain set $\mathcal{D}$. Then, we illustrate the procedure of sampling data points from the domain $e$ by picking the marginal distribution $Pr_\mathbf{X}$ of feature $\mathbf{X}$ while specifying the condition distributions $Pr_Y^e$ of the binary label $Y$ given $\mathbf{X}$

and the domain. Finally, we show that the label space $\mathbf{a} \in \{0,1\}^{|\mathbf{X}|}$, which is the underlying label of $\mathbf{X}$, of $\mathcal{D}$ can be naturally indexed by the vertices of a binary hypercube of dimension $V - 1$.

**Generating $E$ domains from the set $\mathcal{D}$.** As the data generation procedure of o.o.d generalization is of two stages, we show how to sample domains based on the feature space $\mathcal{X}$. The domain-specific label $\mathbf{b}_e$ of the $e$-th domain is generated by the Bernoulli $\left(\frac{1+(2\mathbf{a}-1)h}{2}\right)$ distribution as

$$\Pr(\mathbf{B}_{e,j} = 1) = \begin{cases} \dfrac{1+h}{2}, & \text{if } \mathbf{a}_j = 1; \\ \dfrac{1-h}{2}, & \text{if } \mathbf{a}_j = 0, \end{cases} \quad \forall j \in [V].$$

**Sampling data from $e$-th domain.** Now, $E$ domains are sampled from $\mathcal{D}$ and we present how to sample the training data from the $e$-th domain. Given $p \in [0, 1/(V-1)]$, which will be defined later, $Pr_{\mathbf{X}}^e$ is constructed as follows,

$$\Pr_{\mathbf{X}}^e(\mathbf{X} = \mathbf{x}_j) = \begin{cases} p, & \text{if } 1 \le j \le V - 1; \\ 1 - (V-1)p, & \text{otherwise}. \end{cases}$$

In this way, we ensure that $\Pr_{\mathbf{X}}(\{\mathbf{x}_1, \dots, \mathbf{x}_V\}) = 1$. The labels of the samples follow the distribution $\Pr(Y|\mathbf{X}, \mathbf{B}_e)$ given $\mathbf{B}_e = \mathbf{b}_e \in \{0,1\}^V$ and $\mathbf{X} = \mathbf{x}^e$. Similarly, for a fixed $\mathbf{b}_e \in \{0,1\}^V$, the conditional distribution of $Y^e$ given $\mathbf{x}^e$ is given by two Bernoulli distributions as

$$\text{Bernoulli}\left(\frac{1+(2\mathbf{b}_{e,j}-1)h}{2}\right), \quad \text{if } \mathbf{x}^e = \mathbf{x}_j \text{ and } j \in [V-1];$$
$$\text{Bernoulli}(0), \quad \text{otherwise},$$

where $\mathbf{b}_{e,j}$ is the $j$-th entry of $\mathbf{b}_e$. Thus, we have

$$\eta_{\mathbf{b}_e}(\mathbf{x}) = \Pr(Y = 1|\mathbf{X} = \mathbf{x}, \mathbf{B}_e = \mathbf{b}_e) = \begin{cases} \dfrac{1-h}{2}, & \text{if } \mathbf{x} = \mathbf{x}_j, j \in [V-1] \text{ and } \mathbf{b}_{e,j} = 0; \\ \dfrac{1+h}{2}, & \text{if } \mathbf{x} = \mathbf{x}_j, j \in [V-1] \text{ and } \mathbf{b}_{e,j} = 1; \\ 0, & \text{otherwise}. \end{cases} \quad (6)$$

The corresponding Bayes optimal classifiers on the $e$-th domain and on the mixed data distribution over all domains, denoted by $f_{\mathbf{b}_e}^*$ and $f^*$, respectively, are given by

$$f_{\mathbf{b}_e}^*(\mathbf{x}) = \begin{cases} 0, & \text{if } \mathbf{x} = \mathbf{x}_j, j \in [V-1] \text{ and } \mathbf{b}_{e,j} = 0; \\ 1, & \text{if } \mathbf{x} = \mathbf{x}_j, j \in [V-1] \text{ and } \mathbf{b}_{e,j} = 1; \\ 0, & \text{otherwise}, \end{cases} \quad f^*(\mathbf{x}) = \begin{cases} 0, & \text{if } \mathbf{x} = \mathbf{x}_j, j \in [V-1] \text{ and } \mathbf{a}_j = 0; \\ 1, & \text{if } \mathbf{x} = \mathbf{x}_j, j \in [V-1] \text{ and } \mathbf{a}_j = 1; \\ 0, & \text{otherwise}. \end{cases}$$
$$(7)$$

From each domain, we i.i.d draw $n$ samples. That is, the learning algorithm $\widetilde{f}_{E,n}$ can access to $E \cdot n$ samples in total. Next, we have the following lemma to show that $\mathcal{D} = \{P_{\mathbf{b}_e} : \mathbf{b}_e \in \{0,1\}^V, e \in [E-1]\}$ is in $\mathcal{P}(h, \mathcal{F})$.

**Lemma 6** (Property of "Hard" Case). *All the instances of $\mathcal{D}$ satisfy the Massart noise condition with margin $h$. The distribution $\mathcal{D}$ belongs to $\mathcal{P}(h, \mathcal{F})$, i.e., $\mathcal{D} \in \mathcal{P}(h, \mathcal{F})$.*

Now, the domain distribution $\mathcal{D}$ has been constructed. In the following part, we will show that the problem of learning a classifier in our setting is at least as difficult as recovering the label $\mathbf{a}$. With the Bayes optimal classifier Eq. (7) on the domain set $\mathcal{D}$, we can reduce the problem to a label recovery problem.

### A.3 ANALYZING THE O.O.D. MINIMAX PROBLEM BY REDUCING IT TO A LABEL RECOVERY PROBLEM

We show that the o.o.d minimax problem can be reduced to a label recovery problem by the following theorem.

**Theorem 7** (Reducing to A Label Recovery Problem). *Given a set of domains $\mathcal{D} \in \mathcal{P}(h, \mathcal{F})$ constructed as in Section A.2, then o.o.d minimax problem can be reduced to an estimation problem on a binary hypercube whose vertices index the label space of $\mathcal{D}$, i.e., $\mathbf{a}$, which is the underlying label of $\mathbf{X}$, and the minimax risk $R_{E,n}(h, \mathcal{F})$ (1) satisfies*

$$R_{E,n}(h, \mathcal{F}) \geq \frac{h}{2} \inf_{\tilde{\beta}_{E,n} \in \{0,1\}^{V-1}} \max_{\beta \in \{0,1\}^{V-1}} \mathbf{E}_{P_e \sim \mathcal{D}} \mathbf{E}_{(\mathbf{X}^e, Y^e) \sim P_e} \left\| \tilde{f}^*_{\tilde{\beta}_{E,n}} - f^*_\beta \right\|_{L_1},$$

*where the $L_1$ norm is computed w.r.t the distribution of $\mathbf{X}$, and $\tilde{\beta}$ and $\beta$ are two strings.*

*Proof.* We first apply Lemma 3. Now, the o.o.d generalization minimax risk $R_{E,n}(h, \mathcal{F})$ (1) becomes

$$\inf_{\tilde{f}_{E,n} \in \mathcal{F}} \sup_{\mathcal{D} \in \mathcal{P}(h, \mathcal{F})} \mathbf{E}_{P_e \sim \mathcal{D}} \mathbf{E}_{(\mathbf{X}^e, Y^e) \sim P_e} \left[ L(\tilde{f}_{E,n}) - L(f^*) \right]$$

$$\geq h \inf_{\tilde{f}_{E,n} \in \mathcal{F}} \max_{\beta \in \{0,1\}^{V-1}} \mathbf{E}_{P_e \sim \mathcal{D}} \mathbf{E}_{(\mathbf{X}^e, Y^e) \sim P_e} \left\| \tilde{f}_{E,n} - f^*_\beta \right\|_{L_1},$$

where the $L_1$ norm is w.r.t the distribution of samples $Pr^e_{\mathbf{X}}$, $\tilde{f}_{E,n}$ and $f^*_\beta$ are any classifier in $\mathcal{F}$ and the Bayes optimal classifier trained on the $En$ training samples from $E$ domains, respectively.

Next, by using Lemma 4 with $\tilde{f} = \tilde{f}^*_{E,n}$ and $f = \tilde{f}_{E,n}$, we have

$$h \inf_{\tilde{f}_{E,n} \in \mathcal{F}} \max_{\beta \in \{0,1\}^{V-1}} \mathbf{E}_{P_e \sim \mathcal{D}} \mathbf{E}_{(\mathbf{X}^e, Y^e) \sim P_e} \left\| \tilde{f}_{E,n} - f^*_\beta \right\|_{L_1}$$

$$\geq \frac{h}{2} \inf_{\tilde{\beta}_{E,n} \in \{0,1\}^{V-1}} \max_{\beta \in \{0,1\}^{V-1}} \mathbf{E}_{P_e \sim \mathcal{D}} \mathbf{E}_{(\mathbf{X}^e, Y^e) \sim P_e} \left\| \tilde{f}^*_{\tilde{\beta}_{E,n}} - f^*_\beta \right\|_{L_1},$$

where $\tilde{\beta}_{E,n} = [\tilde{f}^*_{\tilde{\beta}_{E,n}}(\mathbf{x}_1), \ldots, \tilde{f}^*_{\tilde{\beta}_{E,n}}(\mathbf{x}_{V-1})]$ is the binary string that indexes the element of $\{\tilde{f}_{\tilde{\beta}_{E,n}} : \tilde{\beta}_{E,n} \in \{0,1\}^{V-1}\}$. Hence, we have reduced the o.o.d minimax problem to an estimation problem on a binary hypercube (*i.e.*, a label recovery problem). $\square$

By definition, given $n = \infty$, we have the following results

$$\frac{h}{2} \inf_{\tilde{\beta}_{E,\infty} \in \{0,1\}^{V-1}} \max_{\beta \in \{0,1\}^{V-1}} \mathbf{E}_{P_e \sim \mathcal{D}} \mathbf{E}_{(\mathbf{X}^e, Y^e) \sim P_e} \left\| \tilde{f}^*_{\tilde{\beta}_{E,\infty}} - f^*_\beta \right\|_{L_1}$$

$$\geq \frac{h}{2} \inf_{\tilde{\beta}_{E,\infty} \in \{0,1\}^{V-1}} \max_{\beta \in \{0,1\}^{V-1}} \mathbf{E}_{P_e \sim \mathcal{D}} \left\| \tilde{f}^*_{\tilde{\beta}_{E,\infty}} - f^*_\beta \right\|_{L_1}$$

$$= \frac{h}{2} \inf_{\tilde{\beta}_{E,\infty} \in \{0,1\}^{V-1}} \max_{\beta \in \{0,1\}^{V-1}} \mathbf{E}_\beta \left\| \tilde{f}^*_{\tilde{\beta}_{E,\infty}} - f^*_\beta \right\|_{L_1},$$

where $\mathbf{E}_\beta$ represents the expectation with respect to $Pr_\beta$.

Now, we are ready to analyze the $L_1$ norm $\left\| \tilde{f}^*_{\tilde{\beta}_{E,\infty}} - f^*_\beta \right\|_{L_1}$, $\forall \tilde{\beta}_{E,\infty}$, $\beta \in \{0,1\}^{V-1}$. By definition, we derive the following results,

$$\left\| \tilde{f}^*_{\tilde{\beta}_{E,\infty}} - f^*_\beta \right\|_{L_1} = \sum_{j=1}^{V} \Pr_{\mathbf{X}}(\mathbf{X} = \mathbf{x}_j) \left| \tilde{f}^*_{\tilde{\beta}_{E,\infty,j}} - f^*_{\beta_j} \right| = p \sum_{j=1}^{V-1} \left| \tilde{\beta}_{E,\infty,j} - \beta_j \right| = p \cdot d_H \left( \tilde{\beta}_{E,\infty}, \beta \right),$$

where $d_H$ is the Hamming distance, *i.e.*, $d_H(\beta_1, \beta_2) = \sum_j |\beta_{1,j} - \beta_{2,j}|$, and $\beta_{1,j}$ and $\beta_{2,j}$ are the $j$-th items of $\beta_1$ and $\beta_2$. Now, the minimax problem becomes as measuring the distance between two strings $\tilde{\beta}_{E,n}$ and $\beta$,

$$\inf_{\tilde{f}_{E,\infty} \in \mathcal{F}} \sup_{\mathcal{D} \subseteq \mathcal{P}(h, \mathcal{F})} \mathbf{E}_{P_e \sim \mathcal{D}} \mathbf{E}_{(\mathbf{X}^e, Y^e) \sim P_e} \left[ L(\tilde{f}_{E,\infty}) - L(f^*) \right]$$

$$\geq \frac{ph}{2} \inf_{\tilde{\beta}_{E,\infty} \in \{0,1\}^{V-1}} \max_{\beta \in \{0,1\}^{V-1}} \mathbf{E}_\beta d_H \left( \tilde{\beta}_{E,\infty}, \beta \right).$$

To analyze this problem, we apply Theorem 5 and have the following theorem:

**Theorem 8** (Minimax Bound). *Given $H^2\left(Pr_{\tilde{\beta}^*_{E,\infty}}, Pr_\beta\right) \leq \alpha = 2p\left(1 - \sqrt{1-h^2}\right) \leq 2ph^2$, we have the following lower bound for the o.o.d minimax problem,*

$$\inf_{\tilde{f}_{E,\infty} \in \mathcal{F}} \sup_{\mathcal{D} \subseteq \mathcal{P}(h,\mathcal{F})} \mathbf{E}_{P_e \sim \mathcal{D}} \mathbf{E}_{(\mathbf{X}^e, Y^e) \sim P_e} \left[L(\tilde{f}_{E,\infty}) - L(f^*)\right] \geq \frac{V-1}{54Eh},$$

*with $p \in (0, 1/(V-1)]$.*

*Proof.* We need to upper-bound the squared Hellinger distance[2] $H^2\left(Pr_{\tilde{\beta}_{E,\infty}}, Pr_\beta\right), \forall \tilde{\beta}_{E,\infty}, \beta$ that satisfies $d_H\left(\tilde{\beta}_{\infty E,n}, \beta\right) = 1$. Based on the definition of the squared Hellinger distance, we have

$$H^2\left(Pr_{\tilde{\beta}_{E,\infty}}, Pr_\beta\right) = \sum_{i=1}^V \sum_{y \in \{0,1\}} \left(\sqrt{Pr_{\tilde{\beta}_{E,\infty}}(\mathbf{x}_i, b)} - \sqrt{Pr_\beta(\mathbf{x}_i, b)}\right)^2$$

$$= p \sum_{i=1}^{V-1} H^2\left(\text{Bernoulli}\left(\frac{1 + (2\tilde{\beta}_{E,\infty,i} - 1)h}{2}\right), \text{Bernoulli}\left(\frac{1 + (2\beta_i - 1)h}{2}\right)\right).$$

For $j \in [V-1]$, the $j$-th term in the above summation is nonzero if and only if $\tilde{\beta}_{E,\infty,j} \neq \beta_j$, in which case it is equal to the squared Hellinger distance between the Bernoulli $\left(\frac{1-h}{2}\right)$ and Bernoulli $\left(\frac{1+h}{2}\right)$ distributions. Thus,

$$H^2\left(Pr_{\tilde{\beta}_{E,\infty}}, Pr_\beta\right) = p \cdot d_H\left(\tilde{\beta}_{E,\infty}, \beta\right) H^2\left(\text{Bernoulli}\left(\frac{1-h}{2}\right), \text{Bernoulli}\left(\frac{1+h}{2}\right)\right)$$

$$= 2p \cdot d_H\left(\tilde{\beta}_{E,\infty}, \beta\right) \left(\sqrt{\frac{1-h}{2}} - \sqrt{\frac{1+h}{2}}\right)^2$$

$$= 2p \cdot d_H\left(\tilde{\beta}_{E,\infty}, \beta\right) \left(1 - \sqrt{1-h^2}\right).$$

Inserting Theorem 5 with $H^2(Pr_{\tilde{\beta}_{E,\infty}}, Pr_\beta) = 2p\left(1 - \sqrt{1-h^2}\right) = \alpha \leq 2ph^2$, we obtain

$$\inf_{\tilde{f}_{E,\infty}} \sup_{\mathcal{D} \in \mathcal{P}(h,\mathcal{F})} \mathbf{E}_{P_e \sim \mathcal{D}} \mathbf{E}_{(\mathbf{X}^e, Y^e) \sim P_e} \left[L(\tilde{f}_{E,\infty}) - L(f^*)\right]$$

$$\geq \frac{ph}{2} \inf_{\tilde{\beta}_{E,\infty} \in \{0,1\}^{V-1}} \sup_{\beta \in \{0,1\}^{V-1}} \mathbf{E}_\beta d_H(\tilde{\beta}, \beta)$$

$$\geq \frac{ph}{2} \frac{V-1}{2} \left(1 - \sqrt{\alpha E}\right)$$

$$\geq \frac{p(V-1)h}{4} \left(1 - \sqrt{2Eph^2}\right).$$

We let $p = \frac{2}{9h^2 E}$ and now we have

$$\inf_{\tilde{f}_{E,\infty}} \sup_{\mathcal{D} \in \mathcal{P}(h,\mathcal{F})} \mathbf{E}_{P_e \sim \mathcal{D}} \mathbf{E}_{(\mathbf{X}^e, Y^e) \sim P_e} \left[L(\tilde{f}_{E,n}) - L(f^*)\right] \geq \frac{V-1}{54Eh}.$$

$\square$

Next, we discuss the above theorem for different choices of $h$. First, given $h \geq \sqrt{\frac{V-1}{E}}$, we have

$$R_{E,\infty}(\mathcal{F}) \geq \frac{V-1}{54Eh}, \qquad \text{if } h \geq \sqrt{\frac{V-1}{E}}.$$

---

[2]The squared Hellinger distance $H^2(P,Q)$ between $P$ and $Q$ is defined as $H^2(P,Q) = \frac{1}{2}\int_\lambda \left(\sqrt{\frac{dP}{d\lambda}} - \sqrt{\frac{dQ}{d\lambda}}\right)^2 d\lambda.$

When $0 \le h < \sqrt{\frac{V-1}{E}}$, consider $\tilde{h} = \sqrt{\frac{V-1}{E}}$. As $\mathcal{P}(\tilde{h}, \mathcal{F}) \subseteq \mathcal{P}(h, \mathcal{F})$, we have

$$R_{E,\infty}(\mathcal{F}) \ge \frac{V-1}{54E\tilde{h}} = \frac{1}{54}\sqrt{\frac{V-1}{E}}, \qquad \text{if } 0 \le h < \sqrt{\frac{V-1}{E}}.$$

Combine the two cases of $h$. The proof is completed.

# B  PROOFS OF USEFUL LEMMAS AND THEOREMS

## B.1  PROOF OF LEMMA 3

For any classifier $f : \mathcal{X} \to \{0, 1\}$ and any distribution $P$ on $\mathcal{X} \times \{0, 1\}$, we have

$$L(f) - L(f^*) = \mathbf{E}[\mathbf{1}\,(f(\mathbf{X}) \ne Y) - \mathbf{1}\,(f^*(\mathbf{X}) \ne Y)] = \mathbf{E}[|2\eta(\mathbf{X}) - 1||f(\mathbf{X}) - f^*(\mathbf{X})|], \quad (8)$$

where $\mathbf{1}(\cdot)$ is the indicator function. If cond holds, $\mathbf{1}(\text{cond}) = 1$. Otherwise, $\mathbf{1}(\text{cond}) = 0$.

Assuming that $P \in \mathcal{P}(h, \mathcal{F})$, we have

$$L(f) - L(f^*) \ge h\mathbf{E}[|f(\mathbf{x}) - f^*(\mathbf{x})|] = h\|f - f^*\|_{L_1}, \quad (9)$$

where the $L_1$ norm is computed w.r.t the distribution of sampled data $\mathbf{x}$, *i.e.*, $\|f - f^*\|_{L_1} = \sum_{\mathbf{x}} \Pr(\mathbf{X} = \mathbf{x})|f(\mathbf{x}) - f^*(\mathbf{x})|$.

## B.2  PROOF OF LEMMA 4

Let $f$ be any learning algorithm in $\mathcal{F}$, and $\tilde{f}$ be the learning algorithm which is closest to $f_\beta^*$ in $L_1$ norm, *i.e.*, $\tilde{f} = \operatorname{argmin}_{f \in \mathcal{F}, \beta \in \{0,1\}^{V-1}} \|f - f_\beta^*\|_{L_1}$, where $f_\beta^*$ is indexed by $\beta \in \{0, 1\}^{V-1}$. For any $\beta \in \{0, 1\}^{V-1}$, we have

$$\|\tilde{f} - f_\beta^*\|_{L_1} \le \|\tilde{f} - f\|_{L_1} + \|f - f_\beta^*\|_{L_1} \le 2\|f - f_\beta^*\|_{L_1}, \quad (10)$$

where the first one is by the triangle inequality and the second one is due to the definition of $\tilde{f}$.

## B.3  PROOF OF LEMMA 6

First, by Eq. (6), we have $|2\eta_{\mathbf{b}_e}(\mathbf{x}) - 1| \ge h, \forall \mathbf{x} \in \mathcal{X}$. Second, all learning algorithms with VC dimension bigger than $V$ can shatter these data point. There exists at least one $f \in \mathcal{F}$, such that $f_\alpha^*(\mathbf{x}) = f(\mathbf{x})$ for all $\mathbf{x} \in \{\mathbf{x}_1, \ldots, \mathbf{x}_V\}$. Thus, $\mathcal{D} \subseteq \mathcal{P}(h, \mathcal{F})$.

## B.4  PROOF OF THEOREM 5

As the total variation distance can be both upper- and lower-bounded by the Hellinger distance, we have

$$\frac{1}{2}H^2(P, Q) \le \|P - Q\|_{TV} \le H(P, Q). \quad (11)$$

For any $\theta \in \Theta$, let $P_\theta^n$ denote the product of $n$ copies of $P_\theta$, *i.e.*, the joint distribution of $n$ i.i.d samples from $P_\theta$. For any two $\theta, \theta' \in \Theta$ with $d_H(\theta, \theta')$, by letting $P = P_\theta^n$ and $Q = P_{\theta'}^n$, we have the following results

$$\|P_\theta^n - P_{\theta'}^n\|_{TV} \le H(P_\theta^n, Q_\theta^n). \quad (12)$$

Besides, for any $n$ pairs of distributions $(P_{\theta,1}, P_{\theta',1}), \ldots, (P_{\theta,n}, P_{\theta',n})$, where $P_{\theta,*}$ and $P_{\theta',*}$ are copies of $P_\theta$ and $P_{\theta'}$, we have

$$H(P_\theta^n, Q_\theta^n) = H(P_{\theta,1} \times \cdots \times P_{\theta,n}, P_{\theta',1} \times \cdots \times P_{\theta',n}) \le \sqrt{\sum_{i=1}^n H^2(P_{\theta,i}), P_{\theta',i}}. \quad (13)$$

With the assumption (4) on the square of Hellinger, we have

$$\|P_\theta^n - P_{\theta'}^n\|_{TV} \leq \sqrt{\sum_{i=1}^{n} H^2(P_{\theta,i}), P_{\theta',i}} \leq \sqrt{\alpha n}. \tag{14}$$

With Theorem 11, the proof is completed.

## B.5 BACKGROUND AND LEMMAS

We begin this section with some definitions.

The minimax risk $\mathcal{M}(\Theta)$ is defined as

$$\mathcal{M}(\Theta) = \inf_{\hat\theta} \sup_{\theta \in \Theta} \mathbf{E}_\theta[d(\theta, \hat\theta(Z))], \tag{15}$$

where $\Theta$ is a parameter set, $\hat\theta = \hat\theta(Z)$ is an estimator to recover $\theta$ from the observation of a sample $Z$ sampled from an indexed set $\{P_\theta : \theta \in \Theta\}$ of a probability distributions on a finite set $\mathcal{Z}$. $\mathbf{E}_\theta$ represent the expectation with respect to $P_\theta$, *i.e.*,

$$\mathbf{E}_\theta[d(\theta, \hat\theta(Z))] = \sum_{z \in \mathcal{Z}} P_\theta(z) d(\theta, \hat\theta(z)). \tag{16}$$

Besides, the distance metric $d(\cdot, \cdot) : \Theta \times \Theta \to \mathbb{R}^+$ is a pseudometric on $\Theta$ and satisfies the following three properties,

1. Symmetry. $d(\theta, \theta' = d(\theta', \theta)), \forall \theta, \theta' \in \Theta$;

2. Triangle inequality. $d(\theta, \theta') \leq d(\theta, \theta^*) + d(\theta^*, \theta'), \forall \theta, \theta', \theta^* \in \Theta$;

3. Non-negative. $d(\theta, \theta') \geq 0, \forall \theta, \theta' \in \Theta$.

The minimax risk (Eq. (16)) takes the infimum over all estimators $\hat\theta = \hat\theta(Z)$. In another word, this risk tries to find an estimator $\hat\theta$ to minimize the worst-case risk $\sup_{\theta \in \Theta} \mathbf{E}_\theta[d(\theta, \hat(\theta)(Z))]$

We introduce the total variance distance based on the previous definitions.

**Definition 9** (Total Variation Distance). *For any two probability $P, Q \in \mathcal{P}(\mathcal{Z})$, the total variation distance is*

$$\|P - Q\|_{TV} = \frac{1}{2} \sum_{z \in \mathcal{Z}} |P(z) - Q(z)|. \tag{17}$$

*And it can be expressed as follows,*

$$\|P - Q\|_{TV} = 1 - \sum_{z \in \mathcal{Z}} \min(P(z), Q(z)). \tag{18}$$

In the literature, there is an important lemma named two-point method introduced by LeCam (1973) for getting lower bounds on the minimax risk,

**Lemma 10** (LeCam's Lemma). *For any $\theta, \theta' \in \Theta$ and any estimator $\hat\theta$, we have*

$$\mathbf{E}_\theta[d(\theta, \hat\theta(Z))] + \mathbf{E}_\theta[d(\theta', \hat\theta(Z))] \geq d(\theta, \theta') \cdot \sum_{z \in \mathcal{Z}} \min(P_\theta(z), P_{\theta'}(z))$$
$$= d(\theta, \theta')(1 - \|P_\theta - P_{\theta'}\|_{TV}). \tag{19}$$

*Proof.* Given a point $z \in \mathcal{Z}$, assuming $P_\theta(z) \geq P_{\theta'}(z)$, we have

$$\begin{aligned}
& P_\theta(z) d(\theta, \hat\theta(Z)) + P_{\theta'}(z) d(\theta', \hat\theta(Z)) \\
=& P_\theta(z)(d(\theta, \hat\theta(Z)) + d(\theta', \hat\theta(Z))) + (P_{\theta'}(z) - P_\theta(z)) d(\theta', \hat\theta(Z)) \\
\geq& P_\theta(z)(d(\theta, \hat\theta(Z)) + d(\theta', \hat\theta(Z))) \\
\geq& P_\theta(z) d(\theta, \theta').
\end{aligned} \tag{20}$$

Similar, if $P_\theta(z) > P_{\theta'}(z)$, we have

$$P_\theta(z)d(\theta, \hat{\theta}(Z)) \geq P_{\theta'}(z)d(\theta, \theta'). \tag{21}$$

Sum over $\mathcal{Z}$ with the definition of total variation distance. The proof is completed. □

Next, we introduce an important lemma.

**Lemma 11.** *Supposing $\Theta = \{0, 1\}^m$, $\forall \theta, \theta' \in \Theta$, we consider the Hamming metric*

$$d_H(\theta, \theta') = \sum_{i \in [m]} |\theta_i - \theta'_i|, \tag{22}$$

*where $\theta_i$ and $\theta'_i$ are $i$-th entries of $\theta$ and $\theta'$. Then, we can lower-bound the minimax problem as*

$$\mathcal{M} \geq \frac{m}{2} \left( 1 - \max_{d_H(\theta, \theta')=1} \|P_\theta - P_{\theta'}\|_{TV} \right). \tag{23}$$

*Proof.* Let $\pi$ be the uniform distribution on $\Theta = \{0, 1\}^m$ and $\mu_i$ be the joint distribution of a random pair $(\theta, \theta') \in \Theta \times \Theta$, $\forall i \in [m]$, such that the marginal distributions of both $\theta$ and $\theta'$ are equal to $\pi$. Then, the minimax risk can be lower-bounded as,

$$
\begin{aligned}
\mathcal{M}(\Theta) &\geq \inf_{\hat{\theta}} \mathbf{E}_\pi[d(\theta, \hat{\theta}(\mathcal{Z}))] = \inf_{\hat{\theta}} \sum_{i \in [m]} \mathbf{E}_\pi[d(\theta_i, \hat{\theta}_i(\mathcal{Z}))] \\
&\geq \sum_{i \in [m]} \inf_{\hat{\theta}} \mathbf{E}_\pi[d(\theta_i, \hat{\theta}_i(\mathcal{Z}))] \\
&\geq \frac{1}{2} \sum_{i \in [m]} \mathbf{E}_{\mu_i}[d(\theta_i, \theta'_i) \cdot (1 - \|P_\theta - P_{\theta'}\|_{TV})].
\end{aligned}
\tag{24}
$$

The first inequality is due to the supremum over all the prior distribution on $\Theta$ while the third one is by definition and Eq. (18).

Next, since $d(\theta_i, \theta'_i)$, $\forall i \in [m]$, we have

$$
\begin{aligned}
\mathcal{M}(\Theta) &\geq \frac{1}{2} \sum_{i \in [m]} \mathbf{E}_{\mu_i}[d(\theta_i, \theta'_i) \cdot (1 - \|P_\theta - P_{\theta'}\|_{TV})] \\
&\geq \frac{1}{2} \sum_{i \in [m]} \mathbf{E}_{\mu_i}[1 - \|P_\theta - P_{\theta'}\|_{TV}] \\
&\geq \frac{1}{2} \sum_{i \in [m]} \min_{\theta, \theta': d_H(\theta, \theta')=1} [1 - \|P_\theta - P_{\theta'}\|_{TV}] \\
&= \frac{m}{2} (1 - \max_{\theta, \theta': d_H(\theta, \theta')=1} \|P_\theta - P_{\theta'}\|_{TV}).
\end{aligned}
\tag{25}
$$

□

## C EXPERIMENTS

Due to the limitation of space, we present the rest of the experiments in this part.

### C.1 DETAILS OF ALGORITHMS

We include the following algorothms for two multi-domain image classification tasks:

- ERM (Vapnik, 1991) is a famous machine learning algorithm that minimizes the sum of errors across domains and examples.

- IRM (Arjovsky et al., 2019) tries to learn an invariant feature representation $\phi(\cdot)$ to capture the underlying causal mechanism of interest across domains such that the optimal linear classifier on top of that representation matches across domains.

- GroupDRO (Sagawa et al., 2020) learns models minimizing the worst-case training loss over a set of pre-defined groups while increasing the importance of domains with larger errors.

- Mixup (Xu et al., 2020) guarantees domain-invariance in a continuous latent space and guides the domain discriminator in judging samples' difference relative to source and target domains.

- MLDG (Li et al., 2018a) simulates train/test domain shift during training by synthesizing virtual testing domains within each mini-batch.

- CORAL (Sun & Saenko, 2016) aligns correlations of layer activations in deep neural networks to learn domain-invariant features.

- MMD (Li et al., 2018b) extend adversarial autoencoders by imposing the Maximum Mean Discrepancy (MMD (Gretton et al., 2012)) measure to align the distributions among different domains, and matching the aligned distribution to an arbitrary prior distribution via adversarial feature learning.

- DANN (Ganin et al., 2016) encourages the emergence of features that are discriminative for the main learning task on the source domain and indiscriminate with respect to the shift between the domains with an adversarial network.

- C-DANN (Li et al., 2018c) is a variant of DANN matching the conditional distributions on features and labels across domains, for all labels.

## C.2 DETAILS OF MUNIT

Table 4: Details of our MUNIT architecture. We use MUNIT for all the experiments.

| # | Layer |
|---|---|
| 1-3 | Conv2D (in=d, out=64, kernels = $3 \times 3$, padding=1) + ReLU() + GroupNorm(gourps=8) |
| 4-6 | Conv2D (in=64, out=128, kernels = $3 \times 3$, padding=1) + ReLU() + GroupNorm(gourps=8) |
| 7-9 | Conv2D (in=128, out=128, kernels = $3 \times 3$, padding=1) + ReLU() + GroupNorm(gourps=8) |
| 10-12 | Conv2D (in=128, out=128, kernels = $3 \times 3$, padding=1) + ReLU() + GroupNorm(gourps=8) |
| 13 | Global Average-Pooling2D($1 \times 1$) |

## C.3 RESULTS FOR THE TEST-DOMAIN VALIDATION SET (ORACLE) MODEL SELECTION METHOD

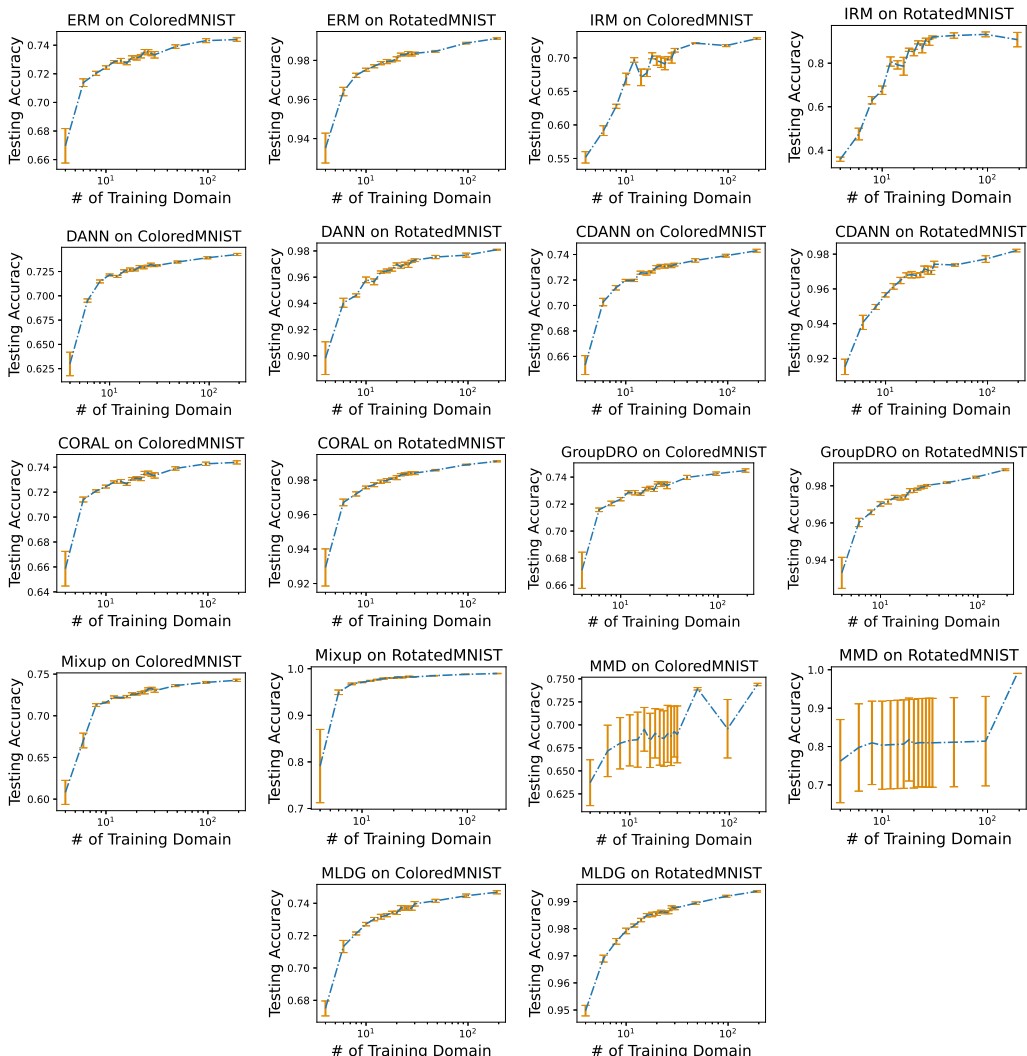

Figure 4: The experimental results on ColoredMNIST and RotatedMINST using ERM (Vapnik, 1991), IRM (Arjovsky et al., 2019), GroupDRO (Sagawa et al., 2020), Mixup (Xu et al., 2020), CORAL (Sun & Saenko, 2016), MMD (Li et al., 2018b), MLDG (Li et al., 2018a), DANN (Ganin et al., 2016) and C-DANN (Li et al., 2018c) w.r.t the number of training domain using the test-domain validation set (oracle) model selection method.

Table 5: The experimental results on **ColoredMNIST** with MMD, DANN and C-DANN w.r.t the number of training domain using the test-domain validation set (oracle) model selection method.

| \# | MMD | DANN | C-DANN |
|---|---|---|---|
| 4 | 0.63715±0.02492 | 0.62982±0.01211 | 0.65319±0.00740 |
| 6 | 0.67170±0.02795 | 0.69513±0.00170 | 0.70268±0.00291 |
| 8 | 0.68003±0.02788 | 0.71480±0.00151 | 0.71387±0.00174 |
| 10 | 0.68314±0.02778 | 0.72152±0.00100 | 0.71965±0.00081 |
| 12 | 0.68391±0.02993 | 0.71994±0.00107 | 0.71965±0.00088 |
| 14 | 0.69516±0.02376 | 0.72537±0.00143 | 0.72573±0.00121 |
| 16 | 0.68314±0.02948 | 0.72692±0.00172 | 0.72525±0.00140 |
| 18 | 0.69082±0.02680 | 0.72663±0.00101 | 0.72637±0.00087 |
| 20 | 0.68658±0.03038 | 0.72984±0.00125 | 0.73000±0.00128 |
| 22 | 0.68514±0.03015 | 0.72891±0.00109 | 0.73129±0.00095 |
| 24 | 0.69025±0.03098 | 0.73077±0.00072 | 0.73061±0.00132 |
| 26 | 0.68784±0.03208 | 0.73238±0.00123 | 0.73113±0.00130 |
| 28 | 0.69278±0.02767 | 0.73090±0.00085 | 0.73084±0.00115 |
| 30 | 0.68967±0.03090 | 0.73126±0.00056 | 0.73206±0.00108 |
| 48 | 0.73923±0.00130 | 0.73479±0.00106 | 0.73534±0.00135 |
| 96 | 0.69581±0.03176 | 0.73919±0.00111 | 0.73906±0.00112 |
| 192 | 0.74392±0.00127 | 0.74257±0.00109 | 0.74295±0.00116 |

Table 6: The experimental results on **RotatedMNIST** with ERM, IRM, GroupDRO, Mixup, MLDG, CORAL, MMD, DANN and C-DANN w.r.t the number of training domain using the test-domain validation set (oracle) model selection method.

| # | ERM | IRM | GroupDRO | Mixup | MLDG | CORAL | MMD | DANN | C-DANN |
|---|---|---|---|---|---|---|---|---|---|
| 4 | 0.93514±0.00764 | 0.35928±0.00964 | 0.93309±0.00837 | 0.79103±0.07864 | 0.94977±0.00195 | 0.92933±0.01077 | 0.76198±0.10829 | 0.89815±0.01249 | 0.91515±0.00440 |
| 6 | 0.96407±0.00215 | 0.47488±0.02682 | 0.96024±0.00223 | 0.94945±0.00486 | 0.96895±0.00127 | 0.96703±0.00193 | 0.79759±0.11377 | 0.94045±0.00343 | 0.94074±0.00407 |
| 8 | 0.97236±0.00106 | 0.62999±0.01684 | 0.96571±0.00121 | 0.96744±0.00181 | 0.97532±0.00105 | 0.97197±0.00123 | 0.80961±0.10859 | 0.94604±0.00121 | 0.94961±0.00144 |
| 10 | 0.97522±0.00078 | 0.67588±0.01914 | 0.97024±0.00121 | 0.97088±0.00078 | 0.97917±0.00102 | 0.97574±0.00078 | 0.80344±0.11473 | 0.95793±0.00214 | 0.95661±0.00125 |
| 12 | 0.97699±0.00077 | 0.80758±0.02122 | 0.97146±0.00127 | 0.97403±0.00100 | 0.98117±0.00056 | 0.97737±0.00120 | 0.80485±0.11497 | 0.95642±0.00223 | 0.96143±0.00161 |
| 14 | 0.97860±0.00082 | 0.79219±0.01893 | 0.97381±0.00103 | 0.97660±0.00077 | 0.98319±0.00062 | 0.97905±0.00095 | 0.80553±0.11508 | 0.96381±0.00093 | 0.96484±0.00180 |
| 16 | 0.97940±0.00107 | 0.78589±0.04100 | 0.97342±0.00112 | 0.97885±0.00120 | 0.98496±0.00061 | 0.97982±0.00110 | 0.80640±0.11523 | 0.96478±0.00143 | 0.96789±0.00136 |
| 18 | 0.97962±0.00041 | 0.86145±0.00724 | 0.97400±0.00117 | 0.98036±0.00050 | 0.98525±0.00082 | 0.98085±0.00044 | 0.81822±0.10803 | 0.96577±0.00121 | 0.96818±0.00186 |
| 20 | 0.98081±0.00104 | 0.84882±0.01469 | 0.97737±0.00110 | 0.98014±0.00089 | 0.98557±0.00067 | 0.98139±0.00115 | 0.80723±0.11537 | 0.96992±0.00096 | 0.96773±0.00128 |
| 22 | 0.98303±0.00067 | 0.89565±0.00707 | 0.97747±0.00104 | 0.98216±0.00087 | 0.98624±0.00056 | 0.98309±0.00063 | 0.80993±0.11512 | 0.96776±0.00097 | 0.96805±0.00135 |
| 24 | 0.98284±0.00055 | 0.86788±0.02032 | 0.97821±0.00057 | 0.98107±0.00086 | 0.98608±0.00056 | 0.98348±0.00083 | 0.81025±0.11498 | 0.97043±0.00098 | 0.97152±0.00162 |
| 26 | 0.98377±0.00077 | 0.90802±0.00669 | 0.9791±0.00078 | 0.98229±0.00075 | 0.98608±0.00056 | 0.98377±0.00073 | 0.80958±0.11576 | 0.96915±0.00173 | 0.97082±0.00229 |
| 28 | 0.98313±0.00087 | 0.89642±0.01438 | 0.97946±0.00074 | 0.98345±0.00059 | 0.98798±0.00046 | 0.98412±0.00079 | 0.81003±0.11583 | 0.97204±0.00081 | 0.96950±0.00116 |
| 30 | 0.98361±0.00059 | 0.92055±0.00437 | 0.98023±0.00049 | 0.98220±0.00077 | 0.98759±0.00059 | 0.98403±0.00085 | 0.80977±0.11579 | 0.97294±0.00102 | 0.97422±0.00169 |
| 48 | 0.98464±0.00044 | 0.92685±0.01240 | 0.98168±0.00044 | 0.98506±0.00056 | 0.98943±0.00044 | 0.98576±0.00041 | 0.81125±0.11603 | 0.97535±0.00123 | 0.97374±0.00069 |
| 96 | 0.98885±0.00040 | 0.93225±0.01117 | 0.98467±0.00052 | 0.98792±0.00046 | 0.99197±0.00037 | 0.98891±0.00029 | 0.81404±0.11650 | 0.97680±0.00153 | 0.97718±0.00191 |
| 192 | 0.99119±0.00038 | 0.90812±0.03302 | 0.98865±0.00048 | 0.98975±0.00039 | 0.99377±0.00027 | 0.99087±0.00033 | 0.99045±0.00026 | 0.98094±0.00034 | 0.98207±0.00074 |

## C.4 RESULTS OF THE TRAINING-DOMAIN VALIDATION SET MODEL SELECTION METHOD

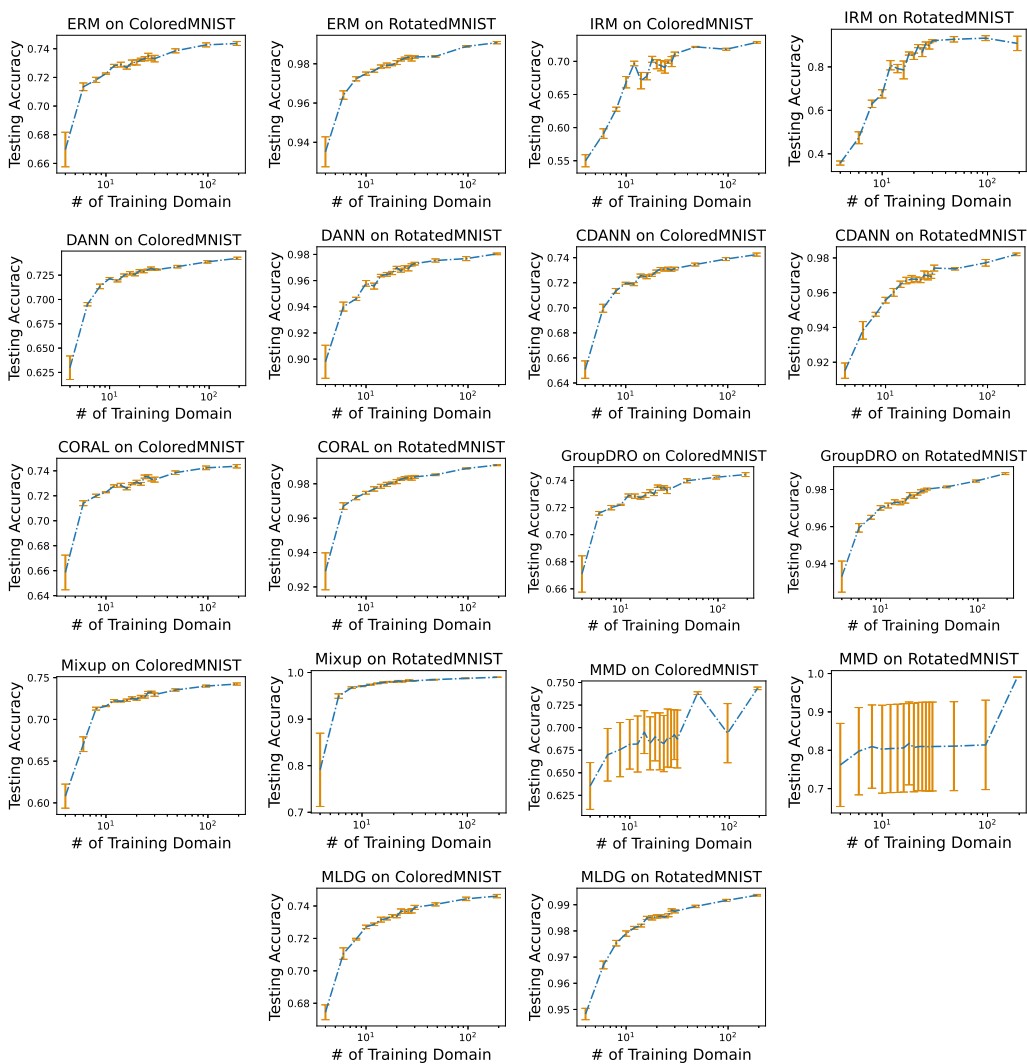

Figure 5: The experimental results on ColoredMNIST and RotatedMINST using ERM (Vapnik, 1991), IRM (Arjovsky et al., 2019), DRO (Sagawa et al., 2020), Mixup (Xu et al., 2020), MLDG (Li et al., 2018a), CORAL (Sun & Saenko, 2016), MMD (Li et al., 2018b), MLDG (Li et al., 2018a), DANN (Ganin et al., 2016) and C-DANN (Li et al., 2018c) w.r.t the number of training domain using the training-domain validation set model selection method.

Table 7: The experimental results on **ColoredMNIST** with MMD, DANN and C-DANN w.r.t the number of training domain using the training-domain validation set model selection method.

| \\# | MMD | DANN | C-DANN |
|---|---|---|---|
| 4 | 0.63538±0.02597 | 0.62982±0.01211 | 0.65068±0.00695 |
| 6 | 0.66993±0.02912 | 0.69510±0.00172 | 0.69963±0.00317 |
| 8 | 0.67565±0.03004 | 0.71342±0.00234 | 0.71345±0.00196 |
| 10 | 0.68154±0.02749 | 0.72152±0.00100 | 0.71956±0.00081 |
| 12 | 0.68186±0.03106 | 0.71911±0.00109 | 0.71891±0.00122 |
| 14 | 0.69503±0.02374 | 0.72483±0.00140 | 0.72573±0.00121 |
| 16 | 0.68263±0.02939 | 0.72692±0.00172 | 0.72476±0.00154 |
| 18 | 0.68983±0.02663 | 0.72563±0.00133 | 0.72598±0.00089 |
| 20 | 0.68481±0.03156 | 0.72949±0.00117 | 0.72901±0.00111 |
| 22 | 0.68234±0.03114 | 0.72862±0.00115 | 0.73109±0.00105 |
| 24 | 0.68848±0.03216 | 0.73055±0.00059 | 0.73061±0.00132 |
| 26 | 0.68784±0.03208 | 0.73190±0.00146 | 0.73113±0.00130 |
| 28 | 0.69221±0.02758 | 0.73087±0.00084 | 0.73003±0.00086 |
| 30 | 0.68739±0.03204 | 0.73068±0.00036 | 0.73132±0.00108 |
| 48 | 0.73826±0.00150 | 0.73373±0.00128 | 0.73453±0.00114 |
| 96 | 0.69388±0.03278 | 0.73868±0.00131 | 0.73890±0.00117 |
| 192 | 0.74353±0.00140 | 0.74228±0.00118 | 0.74247±0.00124 |

Table 8: The experimental results on **RotatedMNIST** with ERM, IRM, GroupDRO, Mixup, MLDG, CORAL, MMD, DANN and C-DANN w.r.t the number of training domain using the training-domain validation set model selection method.

| # | ERM | IRM | GroupDRO | Mixup | MLDG | CORAL | MMD | DANN | C-DANN |
|---|---|---|---|---|---|---|---|---|---|
| 4 | 0.93514±0.00764 | 0.35761±0.00945 | 0.93309±0.00837 | 0.79103±0.07864 | 0.94826±0.00217 | 0.92907±0.01075 | 0.76172±0.10825 | 0.89796±0.01262 | 0.91515±0.00440 |
| 6 | 0.96407±0.00215 | 0.47395±0.02740 | 0.95934±0.00228 | 0.94945±0.00486 | 0.96699±0.00148 | 0.96703±0.00193 | 0.79759±0.11377 | 0.94013±0.00342 | 0.93833±0.00505 |
| 8 | 0.97236±0.00106 | 0.62999±0.01684 | 0.96503±0.00104 | 0.96744±0.00181 | 0.97532±0.00105 | 0.97197±0.00123 | 0.80961±0.10859 | 0.94604±0.00121 | 0.94758±0.00113 |
| 10 | 0.97500±0.00081 | 0.67588±0.01914 | 0.97011±0.00122 | 0.97063±0.00094 | 0.97898±0.00103 | 0.97477±0.00081 | 0.80276±0.11462 | 0.95787±0.00218 | 0.95568±0.00165 |
| 12 | 0.97644±0.00074 | 0.80758±0.02122 | 0.97136±0.00130 | 0.97384±0.00101 | 0.98117±0.00056 | 0.97702±0.00128 | 0.80450±0.11491 | 0.95578±0.00227 | 0.96018±0.00224 |
| 14 | 0.97860±0.00082 | 0.79219±0.01893 | 0.97316±0.00130 | 0.97660±0.00077 | 0.98203±0.00052 | 0.97853±0.00115 | 0.80501±0.11500 | 0.96381±0.00093 | 0.96484±0.00180 |
| 16 | 0.97921±0.00118 | 0.78589±0.04100 | 0.97271±0.00093 | 0.97872±0.00120 | 0.98496±0.00061 | 0.97966±0.00120 | 0.80623±0.11520 | 0.96478±0.00143 | 0.96699±0.00186 |
| 18 | 0.97943±0.00042 | 0.86145±0.00724 | 0.97387±0.00122 | 0.98036±0.00050 | 0.98502±0.00084 | 0.98049±0.00055 | 0.81787±0.10798 | 0.96577±0.00121 | 0.96783±0.00208 |
| 20 | 0.98065±0.00099 | 0.84882±0.01469 | 0.97712±0.00123 | 0.97959±0.00100 | 0.98551±0.00063 | 0.98126±0.00116 | 0.80713±0.11535 | 0.96992±0.00096 | 0.96773±0.00128 |
| 22 | 0.98203±0.00059 | 0.89565±0.00707 | 0.97615±0.00086 | 0.98203±0.00089 | 0.98567±0.0051 | 0.98265±0.0067 | 0.80948±0.11504 | 0.96735±0.0091 | 0.96722±0.00144 |
| 24 | 0.98284±0.00055 | 0.86788±0.02032 | 0.97770±0.00072 | 0.98043±0.00089 | 0.98531±0.00028 | 0.98332±0.00092 | 0.81009±0.11496 | 0.97043±0.00098 | 0.97018±0.00229 |
| 26 | 0.98335±0.00095 | 0.90802±0.00669 | 0.97901±0.00084 | 0.98175±0.00081 | 0.98599±0.00056 | 0.98364±0.00081 | 0.80945±0.11573 | 0.96915±0.00173 | 0.96976±0.00240 |
| 28 | 0.98258±0.00116 | 0.89642±0.01438 | 0.97946±0.00074 | 0.98345±0.00059 | 0.98798±0.00046 | 0.98329±0.00118 | 0.80919±0.11569 | 0.97204±0.00081 | 0.96940±0.00116 |
| 30 | 0.98348±0.00064 | 0.92055±0.00437 | 0.98023±0.00049 | 0.98200±0.00071 | 0.98747±0.00056 | 0.98399±0.00086 | 0.80974±0.11578 | 0.97294±0.00102 | 0.97416±0.00169 |
| 48 | 0.98377±0.00041 | 0.92685±0.01240 | 0.98139±0.00045 | 0.98461±0.00069 | 0.98936±0.00045 | 0.98522±0.00046 | 0.81070±0.11594 | 0.97535±0.00123 | 0.97374±0.00069 |
| 96 | 0.98885±0.00040 | 0.93225±0.01117 | 0.98454±0.00056 | 0.98743±0.00059 | 0.99164±0.00041 | 0.98891±0.00029 | 0.81404±0.11650 | 0.97680±0.00153 | 0.97718±0.00191 |
| 192 | 0.99078±0.00057 | 0.90812±0.03302 | 0.98856±0.00052 | 0.98975±0.00039 | 0.99364±0.00030 | 0.99087±0.00033 | 0.99045±0.00026 | 0.98043±0.00062 | 0.98207±0.00074 |

## C.5 RESULTS OF LEAVE-ONE-DOMAIN-OUT CROSS-VALIDATION METHOD

Table 9: The experimental results on **ColoredMNIST** with ERM, IRM, GroupDRO, Mixup, MLDG, CORAL, MMD, DANN and C-DANN w.r.t the number of training domain with the leave-one-domain-out cross-validation method.

| # | ERM | IRM | GroupDRO | Mixup | MLDG | CORAL | MMD | DANN | C-DANN |
|---|---|---|---|---|---|---|---|---|---|
| 2 | 0.50590±0.00590 | 0.50641±0.00612 | 0.50590±0.00590 | 0.50558±0.00586 | 0.54707±0.01912 | 0.50590±0.00590 | 0.50590±0.00590 | 0.50702±0.00559 | 0.50702±0.00559 |
| 4 | 0.70339±0.00971 | 0.58743±0.02309 | 0.70815±0.00860 | 0.62600±0.01984 | 0.70407±0.01104 | 0.70403±0.00876 | 0.66312±0.07855 | 0.69606±0.00697 | 0.68925±0.01099 |
| 6 | 0.71843±0.00544 | 0.62555±0.01763 | 0.72126±0.00383 | 0.70924±0.00852 | 0.71898±0.01061 | 0.72132±0.00513 | 0.68462±0.06970 | 0.71175±0.00477 | 0.71541±0.00457 |
| 8 | 0.72325±0.00194 | 0.66830±0.01954 | 0.71927±0.00736 | 0.71445±0.00681 | 0.72229±0.00372 | 0.72332±0.00273 | 0.68208±0.08341 | 0.71760±0.00569 | 0.71949±0.00304 |
| 10 | 0.72235±0.00399 | 0.68131±0.01824 | 0.72094±0.00382 | 0.71699±0.00755 | 0.72377±0.00415 | 0.72216±0.00301 | 0.68003±0.08520 | 0.72084±0.00523 | 0.71991±0.00585 |
| 12 | 0.72582±0.00416 | 0.68549±0.02169 | 0.72663±0.00485 | 0.72306±0.00851 | 0.72483±0.00693 | 0.72608±0.00332 | 0.68134±0.08941 | 0.72251±0.00401 | 0.72110±0.00542 |
| 14 | 0.72634±0.00179 | 0.65743±0.02809 | 0.72255±0.00313 | 0.72103±0.00576 | 0.72923±0.00458 | 0.72579±0.00142 | 0.68925±0.07201 | 0.72428±0.00553 | 0.72283±0.00717 |
| 16 | 0.72756±0.00341 | 0.67437±0.03324 | 0.72435±0.00405 | 0.72463±0.00621 | 0.73048±0.00533 | 0.73228±0.00350 | 0.68957±0.08461 | 0.72312±0.00604 | 0.72255±0.00468 |
| 18 | 0.72643±0.00564 | 0.68501±0.02319 | 0.73019±0.00598 | 0.72618±0.00425 | 0.72814±0.00695 | 0.72682±0.00497 | 0.68244±0.08955 | 0.72444±0.00724 | 0.72087±0.00324 |
| 20 | 0.72839±0.00317 | 0.63712±0.00778 | 0.72650±0.00432 | 0.72553±0.00352 | 0.72946±0.00418 | 0.72849±0.00364 | 0.68449±0.08874 | 0.72679±0.00529 | 0.72759±0.00554 |
| 22 | 0.73299±0.00726 | 0.67083±0.02748 | 0.72868±0.00358 | 0.72534±0.00474 | 0.72917±0.00338 | 0.73215±0.00442 | 0.69799±0.07162 | 0.72502±0.00622 | 0.72380±0.00388 |
| 24 | 0.73289±0.00435 | 0.68343±0.03175 | 0.73238±0.00473 | 0.72782±0.00612 | 0.73135±0.00746 | 0.73440±0.00560 | 0.69314±0.08267 | 0.72608±0.00595 | 0.72586±0.00736 |
| 26 | 0.73061±0.00318 | 0.66977±0.02562 | 0.73257±0.00408 | 0.72769±0.01092 | 0.73215±0.00307 | 0.73260±0.00454 | 0.69680±0.06567 | 0.72682±0.00283 | 0.72875±0.00501 |
| 28 | 0.72872±0.00950 | 0.65329±0.03800 | 0.73232±0.00896 | 0.72913±0.00615 | 0.73434±0.00258 | 0.73135±0.00779 | 0.68822±0.08396 | 0.73135±0.00403 | 0.72759±0.00670 |
| 30 | 0.72817±0.00777 | 0.66534±0.03140 | 0.72949±0.00778 | 0.72737±0.00636 | 0.73543±0.00529 | 0.73360±0.00847 | 0.68857±0.08788 | 0.72589±0.00805 | 0.73177±0.00121 |

Table 10: The experimental results on **RotatedMNIST** with ERM, IRM, GroupDRO, Mixup, MLDG, CORAL, MMD, DANN and C-DANN w.r.t the number of training domain with the leave-one-domain-out cross-validation method.

| # | ERM | IRM | GroupDRO | Mixup | MLDG | CORAL | MMD | DANN | C-DANN |
|---|---|---|---|---|---|---|---|---|---|
| 2 | 0.76915±0.02097 | 0.28944±0.04402 | 0.76915±0.02097 | 0.77532±0.01590 | 0.80247±0.02643 | 0.76915±0.02097 | 0.76915±0.02097 | 0.71027±0.04250 | 0.71027±0.04250 |
| 4 | 0.94479±0.00741 | 0.46200±0.07928 | 0.94029±0.01190 | 0.92846±0.01422 | 0.95777±0.00656 | 0.94736±0.00570 | 0.78348±0.32848 | 0.91191±0.01518 | 0.92001±0.01761 |
| 6 | 0.96889±0.00154 | 0.58390±0.06660 | 0.96317±0.00185 | 0.96012±0.00528 | 0.96860±0.00400 | 0.96998±0.00256 | 0.81536±0.31159 | 0.93990±0.00774 | 0.94334±0.00562 |
| 8 | 0.97387±0.00216 | 0.69259±0.03608 | 0.96754±0.00272 | 0.97021±0.00304 | 0.97718±0.00291 | 0.97435±0.00236 | 0.81713±0.31376 | 0.95648±0.00360 | 0.95359±0.00672 |
| 10 | 0.97786±0.00229 | 0.78133±0.06174 | 0.97204±0.00256 | 0.97223±0.00351 | 0.97946±0.00168 | 0.97670±0.00218 | 0.80389±0.34498 | 0.96137±0.00431 | 0.96066±0.00416 |
| 12 | 0.97712±0.00289 | 0.82160±0.03162 | 0.97403±0.00335 | 0.97593±0.00203 | 0.97943±0.00278 | 0.97824±0.00193 | 0.80543±0.34519 | 0.96433±0.00358 | 0.96728±0.00235 |
| 14 | 0.97895±0.00116 | 0.86595±0.01777 | 0.97352±0.00099 | 0.97692±0.00136 | 0.98284±0.00186 | 0.98085±0.00265 | 0.81096±0.33751 | 0.96696±0.00299 | 0.96706±0.00196 |
| 16 | 0.98245±0.00206 | 0.88954±0.02443 | 0.97641±0.00368 | 0.97924±0.00221 | 0.98454±0.00172 | 0.98300±0.00218 | 0.81816±0.32801 | 0.96854±0.00196 | 0.97063±0.00286 |
| 18 | 0.98300±0.00221 | 0.90757±0.01948 | 0.97705±0.00321 | 0.98158±0.00156 | 0.98666±0.00221 | 0.98319±0.00257 | 0.82423±0.31812 | 0.96937±0.00210 | 0.97127±0.00184 |
| 20 | 0.98309±0.00126 | 0.92811±0.01539 | 0.97725±0.00397 | 0.98043±0.00258 | 0.98695±0.00185 | 0.98393±0.00132 | 0.82324±0.31930 | 0.97046±0.00246 | 0.97165±0.00126 |
| 22 | 0.98390±0.00041 | 0.91956±0.02096 | 0.97905±0.00276 | 0.98242±0.00265 | 0.98747±0.00164 | 0.98361±0.00202 | 0.81967±0.32668 | 0.97220±0.00177 | 0.97397±0.00129 |
| 24 | 0.98338±0.00165 | 0.94189±0.01098 | 0.97869±0.00183 | 0.98432±0.00112 | 0.98673±0.00226 | 0.98461±0.00169 | 0.82185±0.32207 | 0.97345±0.00162 | 0.97416±0.00178 |
| 26 | 0.98551±0.00137 | 0.94395±0.01244 | 0.97966±0.00174 | 0.98223±0.00207 | 0.98763±0.00179 | 0.98387±0.00189 | 0.82732±0.31274 | 0.97590±0.00202 | 0.97541±0.00160 |
| 28 | 0.98563±0.00188 | 0.95529±0.00930 | 0.98113±0.00185 | 0.98322±0.00188 | 0.99062±0.00069 | 0.98689±0.00153 | 0.82336±0.32491 | 0.97503±0.00256 | 0.97757±0.00247 |
| 30 | 0.98698±0.00108 | 0.95549±0.00887 | 0.98097±0.00263 | 0.98599±0.00107 | 0.98972±0.00139 | 0.98698±0.00261 | 0.98499±0.00138 | 0.97394±0.00219 | 0.97718±0.00281 |

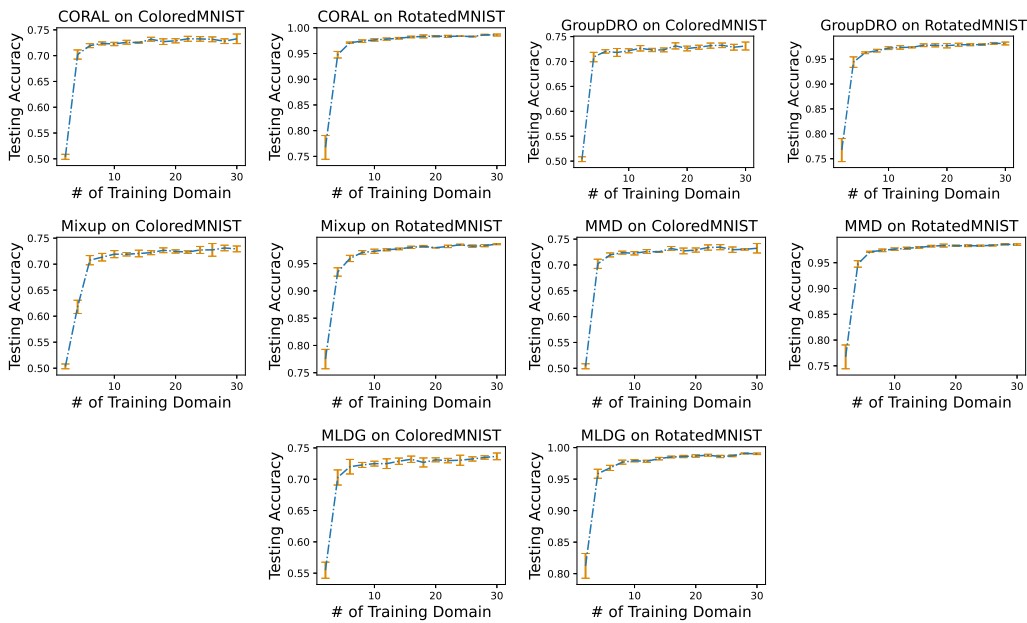

Figure 6: The experimental results on ColoredMNIST and RotatedMINST using DRO (Sagawa et al., 2020), Mixup (Xu et al., 2020), MLDG (Li et al., 2018a), CORAL (Sun & Saenko, 2016), MMD (Li et al., 2018b), and MLDG (Li et al., 2018a) w.r.t the number of training domain with the leave-one-domain-out cross-validation method.

## C.6 RESULTS OF ABLATION STUDY

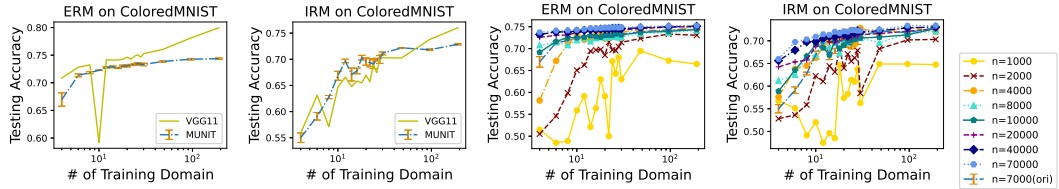

Figure 7: The experimental results on **ColoredMNIST** using ERM (Vapnik, 1991) and IRM (Arjovsky et al., 2019) w.r.t the number of training domain using the training-domain validation set model selection method. The left two figures show the results with different architectures, *i.e.*, MUNIT and VGG11 (Simonyan & Zisserman, 2014), while the left three figures present the corresponding results with different number of $n$.

Table 11: The experimental results on **ColoredMNIST** with ERM w.r.t the number of training domain using the test-domain validation set (oracle) model selection method with changing the number of training images from each domain.

| \# | 1000 | 2000 | 4000 | 7000 (ori) | 8000 | 10000 | 20000 | 40000 | 70000 |
|---|---|---|---|---|---|---|---|---|---|
| 4 | 0.5150 | 0.5050 | 0.5815 | 0.6697 | 0.7080 | 0.6976 | 0.7261 | 0.7324 | 0.7372 |
| 6 | 0.5350 | 0.5463 | 0.6719 | 0.7138 | 0.7146 | 0.7194 | 0.7308 | 0.7381 | 0.7414 |
| 8 | 0.5150 | 0.5988 | 0.7129 | 0.7203 | 0.7133 | 0.7251 | 0.7318 | 0.7392 | 0.7421 |
| 10 | 0.5587 | 0.6506 | 0.7314 | 0.7244 | 0.7194 | 0.7252 | 0.7369 | 0.7399 | 0.7432 |
| 12 | 0.5913 | 0.6625 | 0.7202 | 0.7284 | 0.7241 | 0.7264 | 0.7379 | 0.7417 | 0.7447 |
| 14 | 0.5387 | 0.6944 | 0.7211 | 0.7291 | 0.7237 | 0.7296 | 0.7405 | 0.7435 | 0.7468 |
| 16 | 0.5637 | 0.6969 | 0.7329 | 0.7274 | 0.7237 | 0.7296 | 0.7391 | 0.7434 | 0.7465 |
| 18 | 0.6300 | 0.6981 | 0.7375 | 0.7314 | 0.7261 | 0.7264 | 0.7416 | 0.7447 | 0.7469 |
| 20 | 0.5913 | 0.6850 | 0.7229 | 0.7311 | 0.7267 | 0.7319 | 0.7409 | 0.7449 | 0.7477 |
| 22 | 0.5150 | 0.7150 | 0.7396 | 0.7323 | 0.7271 | 0.7314 | 0.7418 | 0.7455 | 0.7478 |
| 24 | 0.6713 | 0.6969 | 0.7335 | 0.7357 | 0.7294 | 0.7320 | 0.7413 | 0.7463 | 0.7486 |
| 26 | 0.6425 | 0.6956 | 0.7366 | 0.7351 | 0.7281 | 0.7335 | 0.7435 | 0.7463 | 0.7486 |
| 28 | 0.6800 | 0.7044 | 0.7405 | 0.7341 | 0.7300 | 0.7348 | 0.7432 | 0.7465 | 0.7486 |
| 30 | 0.6300 | 0.7144 | 0.7372 | 0.7333 | 0.7270 | 0.7334 | 0.7426 | 0.7464 | 0.7489 |
| 48 | 0.6950 | 0.7231 | 0.7436 | 0.7390 | 0.7347 | 0.7370 | 0.7459 | 0.7477 | 0.7501 |
| 96 | 0.6725 | 0.7331 | 0.7414 | 0.7432 | 0.7376 | 0.7408 | 0.7492 | 0.7499 | 0.7518 |
| 192 | 0.6650 | 0.7306 | 0.7514 | 0.7439 | 0.7419 | 0.7439 | 0.7505 | 0.7512 | 0.7525 |

Table 12: The experimental results on **ColoredMNIST** with IRM w.r.t the number of training domain using the test-domain validation set (oracle) model selection method with changing the number of training images from each domain.

| \# | 1000 | 2000 | 4000 | 7000 (ori) | 8000 | 10000 | 20000 | 40000 | 70000 |
|---|---|---|---|---|---|---|---|---|---|
| 4 | 0.5675 | 0.5519 | 0.5760 | 0.5517 | 0.6206 | 0.5889 | 0.6425 | 0.6562 | 0.6598 |
| 6 | 0.5513 | 0.5363 | 0.6382 | 0.5915 | 0.6253 | 0.6354 | 0.6535 | 0.6793 | 0.6974 |
| 8 | 0.5150 | 0.5587 | 0.6452 | 0.6278 | 0.6647 | 0.6663 | 0.6578 | 0.6976 | 0.7031 |
| 10 | 0.5275 | 0.6225 | 0.6625 | 0.6685 | 0.6684 | 0.6915 | 0.6876 | 0.6987 | 0.7093 |
| 12 | 0.5200 | 0.6106 | 0.6859 | 0.6968 | 0.7020 | 0.6846 | 0.7006 | 0.7049 | 0.7144 |
| 14 | 0.5150 | 0.6444 | 0.6989 | 0.6709 | 0.6864 | 0.6687 | 0.6976 | 0.7089 | 0.7136 |
| 16 | 0.5387 | 0.6312 | 0.7077 | 0.6777 | 0.6961 | 0.6994 | 0.6966 | 0.7086 | 0.7200 |
| 18 | 0.6462 | 0.6825 | 0.7083 | 0.7031 | 0.6737 | 0.6860 | 0.7019 | 0.7146 | 0.7175 |
| 20 | 0.5737 | 0.6400 | 0.6938 | 0.6958 | 0.7000 | 0.6935 | 0.7064 | 0.7148 | 0.7212 |
| 22 | 0.6200 | 0.6706 | 0.6980 | 0.6935 | 0.6964 | 0.7004 | 0.7144 | 0.7101 | 0.7203 |
| 24 | 0.6138 | 0.6550 | 0.7241 | 0.6908 | 0.7024 | 0.7007 | 0.7063 | 0.7183 | 0.7238 |
| 26 | 0.6100 | 0.6625 | 0.6917 | 0.6995 | 0.7121 | 0.7034 | 0.7128 | 0.7177 | 0.7239 |
| 28 | 0.6375 | 0.6787 | 0.7190 | 0.6997 | 0.7067 | 0.7200 | 0.7086 | 0.7224 | 0.7224 |
| 30 | 0.5787 | 0.5844 | 0.7284 | 0.7113 | 0.7063 | 0.7063 | 0.7134 | 0.7190 | 0.7244 |
| 48 | 0.6488 | 0.6825 | 0.7153 | 0.7219 | 0.7016 | 0.7081 | 0.7161 | 0.7221 | 0.7249 |
| 96 | 0.6488 | 0.7019 | 0.7308 | 0.7182 | 0.7157 | 0.7163 | 0.7211 | 0.7286 | 0.7334 |
| 192 | 0.6475 | 0.7031 | 0.7196 | 0.7287 | 0.7213 | 0.7271 | 0.7265 | 0.7310 | 0.7345 |

Table 13: The experimental results on **ColoredMNIST** with ERM w.r.t the number of training domain using the training-domain validation set model selection method with changing the number of training images from each domain.

| \# | 1000 | 2000 | 4000 | 7000 (ori) | 8000 | 10000 | 20000 | 40000 | 70000 |
|---|---|---|---|---|---|---|---|---|---|
| 4 | 0.5150 | 0.5050 | 0.5815 | 0.6697 | 0.7080 | 0.6916 | 0.7258 | 0.7323 | 0.7372 |
| 6 | 0.4850 | 0.5463 | 0.6719 | 0.7135 | 0.7146 | 0.7165 | 0.7303 | 0.7381 | 0.7409 |
| 8 | 0.4888 | 0.5988 | 0.7129 | 0.7183 | 0.7081 | 0.7251 | 0.7311 | 0.7383 | 0.7417 |
| 10 | 0.5587 | 0.6506 | 0.7314 | 0.7226 | 0.7194 | 0.7250 | 0.7369 | 0.7399 | 0.7430 |
| 12 | 0.5913 | 0.6625 | 0.7202 | 0.7280 | 0.7241 | 0.7264 | 0.7379 | 0.7417 | 0.7441 |
| 14 | 0.5212 | 0.6944 | 0.7208 | 0.7289 | 0.7169 | 0.7296 | 0.7405 | 0.7431 | 0.7468 |
| 16 | 0.5637 | 0.6969 | 0.7329 | 0.7268 | 0.7237 | 0.7276 | 0.7391 | 0.7434 | 0.7462 |
| 18 | 0.6300 | 0.6981 | 0.7375 | 0.7304 | 0.7261 | 0.7264 | 0.7409 | 0.7447 | 0.7467 |
| 20 | 0.5913 | 0.6850 | 0.7229 | 0.7305 | 0.7254 | 0.7306 | 0.7409 | 0.7449 | 0.7477 |
| 22 | 0.5000 | 0.7150 | 0.7387 | 0.7323 | 0.7259 | 0.7309 | 0.7418 | 0.7450 | 0.7478 |
| 24 | 0.6713 | 0.6969 | 0.7335 | 0.7330 | 0.7277 | 0.7304 | 0.7413 | 0.7463 | 0.7483 |
| 26 | 0.6425 | 0.6956 | 0.7360 | 0.7350 | 0.7281 | 0.7335 | 0.7435 | 0.7463 | 0.7486 |
| 28 | 0.6800 | 0.7044 | 0.7405 | 0.7336 | 0.7257 | 0.7348 | 0.7432 | 0.7465 | 0.7480 |
| 30 | 0.6300 | 0.7144 | 0.7372 | 0.7331 | 0.7266 | 0.7316 | 0.7426 | 0.7456 | 0.7487 |
| 48 | 0.6950 | 0.7231 | 0.7436 | 0.7386 | 0.7347 | 0.7370 | 0.7459 | 0.7477 | 0.7501 |
| 96 | 0.6725 | 0.7331 | 0.7399 | 0.7427 | 0.7363 | 0.7394 | 0.7492 | 0.7499 | 0.7518 |
| 192 | 0.6650 | 0.7306 | 0.7514 | 0.7437 | 0.7419 | 0.7438 | 0.7505 | 0.7512 | 0.7525 |

Table 14: The experimental results on **ColoredMNIST** with IRM w.r.t the number of training domain using the training-domain validation set model selection method with changing the number of training images from each domain.

| \# | 1000 | 2000 | 4000 | 7000 (ori) | 8000 | 10000 | 20000 | 40000 | 70000 |
|---|---|---|---|---|---|---|---|---|---|
| 4 | 0.5675 | 0.5281 | 0.5760 | 0.5500 | 0.6126 | 0.5889 | 0.6425 | 0.6562 | 0.6598 |
| 6 | 0.5513 | 0.5363 | 0.6382 | 0.5910 | 0.6253 | 0.6354 | 0.6535 | 0.6793 | 0.6974 |
| 8 | 0.4913 | 0.5587 | 0.6452 | 0.6278 | 0.6647 | 0.6663 | 0.6576 | 0.6976 | 0.7031 |
| 10 | 0.5200 | 0.6225 | 0.6625 | 0.6685 | 0.6684 | 0.6915 | 0.6876 | 0.6987 | 0.7093 |
| 12 | 0.4750 | 0.6106 | 0.6859 | 0.6968 | 0.7004 | 0.6846 | 0.7006 | 0.7049 | 0.7144 |
| 14 | 0.4963 | 0.6444 | 0.6989 | 0.6709 | 0.6864 | 0.6687 | 0.6976 | 0.7089 | 0.7134 |
| 16 | 0.4850 | 0.6312 | 0.7077 | 0.6777 | 0.6961 | 0.6994 | 0.6966 | 0.7086 | 0.7200 |
| 18 | 0.6462 | 0.6825 | 0.7083 | 0.7031 | 0.6724 | 0.6830 | 0.7010 | 0.7146 | 0.7167 |
| 20 | 0.5737 | 0.6400 | 0.6938 | 0.6957 | 0.7000 | 0.6904 | 0.7064 | 0.7148 | 0.7212 |
| 22 | 0.5825 | 0.6706 | 0.6944 | 0.6935 | 0.6964 | 0.7004 | 0.7143 | 0.7101 | 0.7203 |
| 24 | 0.6138 | 0.6550 | 0.7241 | 0.6908 | 0.7024 | 0.7007 | 0.7063 | 0.7183 | 0.7238 |
| 26 | 0.6100 | 0.6625 | 0.6917 | 0.6995 | 0.7121 | 0.7034 | 0.7128 | 0.7177 | 0.7239 |
| 28 | 0.6375 | 0.6787 | 0.7190 | 0.6997 | 0.7067 | 0.7200 | 0.7086 | 0.7224 | 0.7224 |
| 30 | 0.5625 | 0.5844 | 0.7284 | 0.7113 | 0.7063 | 0.7063 | 0.7134 | 0.7190 | 0.7226 |
| 48 | 0.6488 | 0.6825 | 0.7153 | 0.7219 | 0.6959 | 0.7081 | 0.7160 | 0.7221 | 0.7249 |
| 96 | 0.6488 | 0.7019 | 0.7308 | 0.7182 | 0.7157 | 0.7133 | 0.7211 | 0.7285 | 0.7334 |
| 192 | 0.6475 | 0.7031 | 0.7196 | 0.7287 | 0.7213 | 0.7271 | 0.7265 | 0.7306 | 0.7335 |

Table 15: The experimental results on **ColoredMNIST** with ERM w.r.t the number of training domain using the training-domain validation set model selection method with MUNIT and VGG11.

| \\# | 4 | 6 | 8 | 10 | 12 | 14 | 16 | 18 | 20 | 22 | 24 | 26 | 28 | 30 | 48 | 96 | 192 |
|---|---|---|---|---|---|---|---|---|---|---|---|---|---|---|---|---|---|
| MUNIT | 0.6697 | 0.7135 | 0.7183 | 0.7226 | 0.7280 | 0.7289 | 0.7268 | 0.7304 | 0.7305 | 0.7323 | 0.7330 | 0.7350 | 0.7336 | 0.7331 | 0.7386 | 0.7427 | 0.7437 |
| VGG11 | 0.7085 | 0.7281 | 0.7324 | 0.5918 | 0.7417 | 0.7412 | 0.7413 | 0.7426 | 0.7464 | 0.7428 | 0.7481 | 0.7507 | 0.7465 | 0.7527 | 0.7603 | 0.7810 | 0.7997 |

Table 16: The experimental results on **ColoredMNIST** with IRM w.r.t the number of training domain using the training-domain validation set model selection method with MUNIT and VGG11.

| \\# | 4 | 6 | 8 | 10 | 12 | 14 | 16 | 18 | 20 | 22 | 24 | 26 | 28 | 30 | 48 | 96 | 192 |
|---|---|---|---|---|---|---|---|---|---|---|---|---|---|---|---|---|---|
| MUNIT | 0.5500 | 0.5910 | 0.6278 | 0.6685 | 0.6968 | 0.6709 | 0.6777 | 0.7031 | 0.6957 | 0.6935 | 0.6908 | 0.6995 | 0.6997 | 0.7113 | 0.7219 | 0.7182 | 0.7287 |
| VGG11 | 0.5589 | 0.6312 | 0.5716 | 0.6466 | 0.6659 | 0.6476 | 0.6585 | 0.6548 | 0.6694 | 0.6571 | 0.6897 | 0.6743 | 0.6994 | 0.7029 | 0.7025 | 0.7379 | 0.7603 |

Table 17: The experimental results on **ColoredMNIST** with ERM w.r.t the number of training domain using the test-domain validation set (oracle) model selection method with MUNIT and VGG11.

| \\# | 4 | 6 | 8 | 10 | 12 | 14 | 16 | 18 | 20 | 22 | 24 | 26 | 28 | 30 | 48 | 96 | 192 |
|---|---|---|---|---|---|---|---|---|---|---|---|---|---|---|---|---|---|
| MUNIT | 0.6697 | 0.7138 | 0.7203 | 0.7244 | 0.7284 | 0.7291 | 0.7274 | 0.7314 | 0.7311 | 0.7323 | 0.7357 | 0.7351 | 0.7341 | 0.7333 | 0.7390 | 0.7432 | 0.7439 |
| VGG11 | 0.7087 | 0.7281 | 0.7324 | 0.5918 | 0.7417 | 0.7412 | 0.7442 | 0.7429 | 0.7464 | 0.7456 | 0.7489 | 0.7507 | 0.7510 | 0.7553 | 0.7605 | 0.7810 | 0.7997 |

Table 18: The experimental results on **ColoredMNIST** with IRM w.r.t the number of training domain using the test-domain validation set (oracle) model selection method with MUNIT and VGG11.

| \\# | 4 | 6 | 8 | 10 | 12 | 14 | 16 | 18 | 20 | 22 | 24 | 26 | 28 | 30 | 48 | 96 | 192 |
|---|---|---|---|---|---|---|---|---|---|---|---|---|---|---|---|---|---|
| MUNIT | 0.5517 | 0.5915 | 0.6278 | 0.6685 | 0.6968 | 0.6709 | 0.6777 | 0.7031 | 0.6958 | 0.6935 | 0.6908 | 0.6995 | 0.6997 | 0.7113 | 0.7219 | 0.7182 | 0.7287 |
| VGG11 | 0.5589 | 0.6312 | 0.5803 | 0.6466 | 0.6659 | 0.6476 | 0.6608 | 0.6548 | 0.6694 | 0.6571 | 0.6897 | 0.6743 | 0.6994 | 0.7036 | 0.7025 | 0.7382 | 0.7603 |

