# OpenReview forum: "Lost Domain Generalization Is a Natural Consequence of Lack of Training Domains"
_ICLR.cc/2023/Conference — Submitted to ICLR 2023_

### Official Review · Reviewer_8Svr · 2022-10-23

**Confidence:** 4
**Correctness:** 2
**Technical Novelty And Significance:** 2
**Empirical Novelty And Significance:** 2
**Recommendation:** 3

**Clarity, Quality, Novelty And Reproducibility:**

**Clarity.**  The writing is acceptable, but there are various grammatical mistakes throughout.  The more challenging aspect of the paper is the difficulty of understanding construction needed for Thm. 1.

**Quality.**  I skimmed the proofs in the appendix, and they seem rigorous.

**Novelty.**  As far as I know, this setting and perspective has not been studied in the literature.  There are aspects of novelty in trying to prove lower bounds for DG.  This is a strong point of the paper.

**Reproducibility.**  The experiments seem relatively reproducible, given that they are built on DomainBed.

**Strength And Weaknesses:**

### Strengths

**Approach to DG.**  In some sense, this is an insightful way of thinking about DG in my opinion.  To elaborate, in general, DG is impossible, given that the training and test domains could be arbitrarily different.  In this way, lower bounds can provide concrete settings where DG is impossible, which may give us insights into when DG is theoretically possible.  I think the direction of this paper is promising for this reason -- it seeks to give a simple, theoretical setting which can shed light on the problem.

### Weaknesses

**Problem setup.**  Section 3 is not written in an accessible way.  In this sentence:

> "We will use $L(f)$ to represent the expected 0-1 loss of classifier f w.r.t. the mixture of data distributions of all domains"

What do the authors mean by "the mixture of data distributions of all domains?"  The notion of "domains" has not been introduced yet in this paper.  And even if they had, the authors do not define what they mean by "mixture."  The authors then introduce $\mathcal{P}$ as the "distribution of distribution."  Having read the paper several times over, I'm still not sure what this means.  Are the authors making an assumption that the domains are drawn from a probability distribution over domains?  Should it be obvious what the "domain distribution" is without defining it?

Furthermore, I don't understand the relationship between $(X,y)$ and $(x^e,y^e)$.  $(X,y)$ take on the distribution over the "mixture of data distributions" whereas $(x^e,y^e)$ are defined with respect to a particular distribution.  So again, the question is, how does the mixing work WRT $P^e$ (the distribution over the $e$th domain's data)?

It's also unclear why the authors define $\mathcal{P}$ in the way that they do in the "Problem setups" section.  One point of confusion is that $\mathcal{P}$ is referred to as a "domain distribution," whereas earlier $P^e$ was referred to as "the data distribution of the $e$-th domain."  So from the perspective of the reader, it's now unclear whether $\mathcal{P}$ is a distribution over data in a particular domain or the distribution of data in a particular domain.  The "definition" of $\mathcal{P}$ does not clear this up, as $\mathcal{P}(h,\mathcal{F})$ reflexively uses $\mathcal{P}$ without defining it.   Furthermore, from the definition it's not clear how $\mathcal{P}(h,\mathcal{F})$ depends on $\mathcal{F}$, or whether it depends on $\mathcal{F}$ at all -- $\mathcal{F}$ never appears in the definition.  A similar point could be made regarding how $h$ relates to $\mathcal{P}$.  If $\mathcal{P}$ is not a distribution over data in a particular domain (as is suggested in various places and contradicted in others), then how id $\mathcal{P}$ related to $h$.  If, on the other hand, $\mathcal{P}$ is a domain distribution, what is the relevance of $P^e$?

Putting all this aside, it's also unclear why the authors consider this model.  What makes this Bernoulli setting interesting.  The authors are interested in showing impossibility for domain generalization.  Therefore, one would assume that they would want to convince the reader that their model is natural and flexible, in the sense that this model is one in which we might hope that DG would be theoretically possible.  However, no such discussion is given.  Without convincing the reader that this is the right model to study, it's much harder for the reader to get a sense for why this result makes a contribution.  To this end, a strong point of reference is the paper *The Risks of Invariant Risk Minimization* (https://arxiv.org/pdf/2010.05761.pdf), in which the authors prove impossibility results for IRM.  This paper has gained traction in part due to the fact that the Gaussian model they consider is natural -- meaning that it is one in which one would hope DG would be possible.  The authors spend a considerable amount of the preliminaries section discussing why this model is natural, and they propose various extensions to validate this.  I think that adding a similar discussion to this paper would strengthen the contribution.

The problem setting would be clearer if the authors could define what they mean by the Bayes optimal classifier in this setting. Is the Bayes optimal classifier defined WRT the average over some family of distributions?  Is it defined as the classifier that can do well on domains in the minimax sense (vis-a-vis (1))?  Without knowing this, it's hard to judge the significance of the results.  Relatedly, it's not clear how $E$ and $n$ factor into (1).  The optimization problem in (1) is a population-level problem.  If the authors have another problem in mind wherein there are only finitely many domains and finitely many data points sampled per domain, they should state this problem and discuss its relationship to (1).

Other questions:

- What does it mean for $x$ to be the "implementation" of $X$?  Does this mean that $x\sim X$?
- What do the authors mean when they call this setting "our hard instance?"  What is hard about it?  Is this in contrast to an easier instance?

**Implications of the lack of preliminaries on understanding Thm. 1.**  As described above, I expect that many readers would be confused after reading the preliminaries in Section 3.  The result of this is that it's difficult to grok Thm. 1.  For instance, it's unclear what it means for $n=\infty$ vis-a-vis eq. (1).  As $n$ does not seen to play an explicit role in this problem, there's no way to interpret $n=\infty$.  Furthermore, as the Bayes optimal classifier has not been defined in this domain generalization setting, it's unclear what (2) means.

Furthermore, it's not clear what is "information theoretic" about this bound, as mentioned by the authors.  Is this due to the proof technique, i.e. were techniques from information theory used to prove the result?

The authors say that their result stands in contrast to the results of DomainBed, where few training domains are available.  I think that this statement is not necessarily true.  Consider the sentence:

> "Theorem 1 predicts the failures of future algorithms on these datasets and attributes the poor performance of existing o.o.d. algorithms to the lack of training domains."

This would only be true if the theory directly applies to DomainBed.  Clearly, as the domains are not generated by this Bernoulli model, the lower bound in Thm. 1 does not hold for datasets like ColoredMNIST.  A more nuanced analysis of the implications of this result is needed, rather than claiming that this bound uniformly explains away the poor performance of algorithms used on DomainBed.

**Comparison to past work.**  The authors discuss the differences between their work and that of another paper in Section 3.  However, due to the lack of details given about their approach, it's hard to understand these differences.  For instance, the authors say that they have a "two-stage data generative process."  Why is this an important difference?  What was done in past work?  What is the so-called "supporting space" and how is it relevant to thsi problem?

**Experimental presentation.**  It would seem necessary to disentangle the effect of having finitely many data points from having finitely many domains in these experiments as well.  Could it not be the case that test accuracy rises given more domains simply because the classifier has more data points to train on.  It would seem necessary to try an experiment where the total number of data points is fixed, and so as the classifier has access to more domains, it simultaneously has access to fewer samples per domain.

How did the authors pick the number 192 -- the maximum number of training domains?

The authors show that the test performance increases as a function of the number of domains used at training time.  Indeed, on ColoredMNIST, the authors achieve results near to the optimal score of 75% (given that there is label-flipping) in Table 2.  This indicates that given only ~200 training domains, DG is possible in this setting.  This seems to violate the theory, because Thm. 1 indicates that $E$ must be larger than the VC dimension of $\mathcal{F}$.  As ColoredMNIST uses neural networks, the VC dimension is rather larger here -- certainly larger than 192 given that a linear classifiers would have a VC-dimension of 784+1 for MNIST, and NNs should have a larger VC-dimension than do linear classifiers.  Of course, one could argue that the setting of the theorem does not apply to Colored/Rotated-MNIST.  However, if this argument is used, then it begets the question: How are these experiments serving to reinforce the message given in Thm. 1?


**Summary Of The Paper:**

This paper asks the question: Is the poor performance of domain generalization (DG) algorithms a consequence of not having access to sufficiently many training domains?  The authors prove a lower bound which suggests that the answer to this question is "yes."  They then provide experiments meant to reinforce this theory.

**Summary Of The Review:**

In summary, this paper takes an interesting perspective on DG.  Namely, it seeks to show impossibility results based on the number of training domains.  Unfortunately, I found it quite challenging to read this paper given that the construction used in the main result is not described in sufficient detail in the main text.  Without defining key quantities, e.g. the domain generalization Bayes optimal classifier and the dependence on $E$ and $n$, I find it rather challenging to interpret the main result.  I also feel that the experiments have the following flaw: it's either the case that (a) the experiments invalidate the theory, as strong performance is achieved with fewer domains than the VC-dimension of $\mathcal{F}$, or (b) the assumptions of the theorem do not apply in the experiments, in which case the experiments have no real bearing on the contribution of this paper.  For these reasons, while this paper makes some interesting points, I am going to recommend that this paper not be accepted.

---

### Official Review · Reviewer_qD5b · 2022-10-24

**Confidence:** 4
**Correctness:** 4
**Technical Novelty And Significance:** 4
**Empirical Novelty And Significance:** 4
**Recommendation:** 6

**Clarity, Quality, Novelty And Reproducibility:**

The results are novel and likely to have a positive impact on the domain generalisation field. The particular technique used in proving the main theoretical result is similar to Massart and Nedelec (2006b), but I think this is nothing to be concerned about; the focus of the paper is not on developing a new proof technique, but on demonstrating a novel discovery about the domain generalisation problem.

**Strength And Weaknesses:**

### Strengths
* Given that a lot of papers in the DG community make dubious claims about the efficacy of their methods, investigation into fundamental hardness of DG is a very timely and significant research direction. To my knowledge, this is the first such paper that provides this kind of hardness result in a DG setting.
* The result it quite intuitive and provides further support from a different perspective to a quite new thread on theoretical DG research pointing to model complexity being a significant factor in determining the o.o.d. performance of a model.
* There is good discussion of prior work on theoretical analysis of domain generalisation, and the related work on constructing minimax bounds using similar techniques.
* The experimental results provide a nice corroboration of the theoretical findings.

### Weaknesses
* The paper could use a bit more in the way of a problem definition; a more explicit definition of $\mathcal{P}$ and $P_e$ would make the paper much more readable. The current problem setup jumps immediately into defining the specific form of distribution used in the process of proving the minimax bound. While it is useful to introduce this family of distributions at some point, I think it would be good to first set the scene a bit by introducing the form of bound that will be proved first. It would then make sense to introduce the hard instance, mentioning that it is of course contained in the set of all possible instances that the bound maximises over.
* The presentation of the experimental results is not very easy to understand. It would be better if the domain-efficiency measurements were not reported in tables or many plots that each present a single dataset-method combination. Instead, it would be both more compact and easier to interpret if the tables were discarded and the results summarised by constructing a single plot for each dataset, with multiple lines---one for each method.


**Summary Of The Paper:**

The paper undertakes a minimax analysis of a formulation of the domain generalisation problem. When considering an optimistic case where one has access to infinite data from a finite number of domains, it is determined that the number of required domains to obtain a solution within performance of the Bayes optimal classifier scales as $poly(1/\epsilon)$. In addition, the paper includes an empirical investigation into how well existing algorithms scale with the number of training domains, and it is shown that the empirical behaviour agrees with the theoretical analysis.

**Summary Of The Review:**

Both the theoretical and empirical contributions make this a good paper, but the presentation of the experimental results could be improved.

I will increase my score to 8 if all weaknesses are addressed.

---

### Official Review · Reviewer_2RNe · 2022-10-25

**Confidence:** 3
**Correctness:** 3
**Technical Novelty And Significance:** 2
**Empirical Novelty And Significance:** 2
**Recommendation:** 3

**Clarity, Quality, Novelty And Reproducibility:**

This work generalizes Massart & Nédélec (2006b) to OOD generalization, and the novelty needs  to be further justified.

**Strength And Weaknesses:**


### Strength
- The authors provide theoretical and empirical evidence to justify their claims.

### Weaknesses
- Main assumptions made are not clearly stated. It is not clear what kind of properties of the DG problem you have used for deriving the theory (Thm. 1).  It would nice if the authors could re-organize the writtings, especially the proof techniques, so that one can follow the details in an easier way.
-  There have been plenty of works on analyzing the domain complexity (or related ones) for DG or unsupervised domain adaption, to name a few [Zhao et al. 2019, Wang et al. 2022, Chen et al. 2021].   They have derived bounds by taking into account refined properties of distribution shift problem being studied. For example, the discrepancy between source and target domains, difference of the underlying labeling processed et al.  This kinds of “refined” conclusions would benefit more for studying the DG/DA problems. So would it be possible to improve your conclusion by considering more refined properties of DG?  If so, what would you do then?

- Tightness of the bound in Thm. 1.     The tightness of the bound matters a lot. Have you studied the tightness by considering some well-designed problem instances of DG?   This might also be related to the above concerns, if more refined assumptions are taken into account, the bound would become sharper.

- More discussions with related works are needed to understand/justify  the main theoretical result of this paper.
First, the original IRM settting [Arjovsky et al., 2019], including those who argue about the drawbacks of IRM [Kamath et al, 2021; Rosenfeld et al., 2021; Ahuja et al., 2021] all obey the general definitions of OOD generalization (thus the settings in this paper as well). However, they generally have results that prove IRM is sufficient to find the underlying optimal classifier, given certain a mild number of environments (e.g., linear w.r.t. the number of feature dimensions). These conclusions do not appear to be in line with the conclusion in this paper. Can you explain about the implications of this observation?

### References

Zhao, H., Des Combes, R. T., Zhang, K., & Gordon, G. (2019, May). On learning invariant representations for domain adaptation. In International Conference on Machine Learning (pp. 7523-7532). PMLR.

Wang, H., Si, H., Li, B., & Zhao, H. (2022). Provable Domain Generalization via Invariant-Feature Subspace Recovery. arXiv preprint arXiv:2201.12919.

Chen, Y., Rosenfeld, E., Sellke, M., Ma, T., & Risteski, A. (2021). Iterative feature matching: Toward provable domain generalization with logarithmic environments. arXiv preprint arXiv:2106.09913.

Kamath et al., Does Invariant Risk Minimization Capture Invariance? AISTATS 2021.

Rosenfeld et al., The Risks of Invariant Risk Minimization. ICLR 2021.

Ahuja et al., Invariance Principle Meets Information Bottleneck for Out-of-Distribution Generalization. NeurIPS 2021.


**Summary Of The Paper:**

This paper argues the existing drawback of OOD learning algorithms is because of lacking sufficient domains. The authors provide a hardness result that all learning algorithms require at least $poly (1/\epsilon)$ number of training domains to achieve an $\epsilon$ excess error.

**Summary Of The Review:**

See the comments above

---

### Decision · Program_Chairs · 2023-01-20

**Decision:**

Reject

**Justification For Why Not Higher Score:**

The paper requires a major revision and thereby another round of peer review.

**Justification For Why Not Lower Score:**

N/A

**Metareview: Summary, Strengths And Weaknesses:**

The paper provides a hardness result in terms of the lower bound for the domain generalization (DG) problem, which is an important problem. However, there is a consensus among the reviewers:  All reviewers raised the poor presentation as a major concern, especially since the problem setup is ambiguous (Reviewers 2RNe, qD5b, 8Svr). The authors did not provide any responses to clarify this issue. As a result, I cannot recommend this paper in its current form for publication at ICLR 2023.

**Summary Of Ac-Reviewer Meeting:**

N/A